# Validation of the Unesp-Botucatu composite scale to assess acute postoperative abdominal pain in sheep (USAPS)

Nuno Emanuel Oliveira Figueiredo Silva[1][☉], Pedro Henrique Esteves Trindade[1][☉], Alice Rodrigues Oliveira[1][‡], Marilda Onghero Taffarel[2][‡], Maria Alice Pires Moreira[3][‡], Renan Denadai[1][‡], Paula Barreto Rocha[1][‡], Stelio Pacca Loureiro Luna[1][☉]*

1 Department of Veterinary Surgery and Animal Reproduction, School of Veterinary Medicine and Animal Science, São Paulo State University (Unesp), Botucatu, São Paulo, Brazil, 2 Department of Veterinary Medicine, State University of Maringá, Umuarama, Paraná, Brazil, 3 Goiano Federal Institute, Urutaí Campus, Department of Veterinary Medicine, Urutaí, GO, Brazil

☉ These authors contributed equally to this work.
‡ These authors also contributed equally to this work.
* stelio.pacca@unesp.br

**Data Availability Statement:** All relevant data are within the manuscript and its Supporting Information files.

**Funding:** SPLL - Financial support from the São Paulo Research Foundation – FAPESP (Process

## Abstract

A scale with robust statistical validation is essential to diagnose pain and improve decision making for analgesia. This blind, randomised, prospective and opportunist study aimed to develop an ethogram to evaluate behaviour and validate a scale to assess acute ovine post-operative pain. Elective laparoscopy was performed in 48 healthy sheep, filmed at one pre-operative and three postoperative moments, before and after rescue analgesia and 24 hours after. The videos were randomised and assessed twice by four evaluators, with a one-month interval between evaluations. Statistical analysis was performed using R software and differences were considered significant when p <0.05. Based on the multiple association, a unidimensional scale was adopted. The intra- and inter-observer reliability ranged from moderate to very good (intraclass correlation coefficient ≥ 0.53). The scale presented Spearman correlations > 0.80 with the numerical, simple descriptive, and visual analogue scales, and a correlation of 0.48 with the facial expression scale. According to the mixed linear model, the scale was responsive, due to the increase and decrease in pain scores of all items after surgery and analgesic intervention, respectively. All items on the scale demonstrated an acceptable Spearman item-total correlation (0.56–0.76), except for appetite (0.25). The internal consistency was excellent (Cronbach's α = 0.81) and all items presented specificity > 0.72 and sensitivity between 0.61–0.90, except for appetite. According to the Youden index, the cut-off point was ≥ 4 out of 12, with a diagnostic uncertainty zone of 4 to 5. The area under the curve > 0.95 demonstrated the excellent discriminatory capacity of the instrument. In conclusion, the Unesp-Botucatu pain scale in sheep submitted to laparoscopy is valid, reliable, specific, sensitive, with excellent internal consistency, accuracy, discriminatory capacity, and a defined cut-off point.

2010/08967-0 and 2017/12815-0). http://www.
fapesp.br/en/. NEOFS - Grant from Coordenação de
Aperfeiçoamento de Pessoal de Nível Superior –
Brazil (CAPES). The funders had no role in study
design, data collection and analysis, decision to
publish, or preparation of the manuscript.

**Competing interests:** The authors have declared
that no competing interests exist.

## Introduction

The lack of valid and reliable instruments to recognise and quantify pain in farm animals compromises their welfare state and limits the use of analgesics in these species [1–4]. Sheep are subjected to several surgical painful procedures often without appropriate use of anaesthesia or analgesia [5–7]. The sheep species is an experimental model for humans due to its similarity in size and weight. In 2014 60,209 sheep were used in research in the European Union, an increase of 108% compared to 2011. Pigs followed by sheep are more commonly used than dogs and non-human primates as non-rodent models for research and teaching [8,9].

Although there are several experimental methods to assess nociception [10–14] they are unreliable and difficult to use in unhabituated clinical patients. Actigraphy can be used to monitor sheep activity from a distance, however, this method requires specific equipment [15]. Other physiological measures, such as hair cortisone concentration [16], heart rate variability [17], blood pressure, ocular and rectal temperature, electromyography, and electroencephalography are not clinically feasible and some require physical restraint [18].

The behavioural expression of pain replaces the absence of verbal expression of the animals. Behaviour is easy to observe, does not require equipment and restraint of the animal, does not generate stress, and has no cost, thus being applicable both clinically and experimentally [19].

In contrast to cattle and goats, when sheep suffer pain under restraint they tend to remain more silent [20] and only express pain behaviours when released [21]. These behaviours are: reduced interaction with the environment and with other animals, gait abnormalities, lameness, stamping feet on the ground, turning of the head, hyporexia, abnormal vocalisation, lip-licking movement, curved lips, gnashing of teeth, tremors, and strong tail wagging [5,20–22].

Two analyses are essential to develop and validate a pain scale: validity indicates whether the instrument effectively measures the attribute which it was designed for [23], and reliability guarantees equivalence of results when the measure is evaluated by the same observer on different occasions or by different observers on the same occasion [24]. The scale must also be responsive; scores should increase after a painful stimulus and reduce after analgesia [25]. Behaviour-based pain scales have been developed in dogs [23,26–28], cats [25,29], horses [24,30,31], cattle [19], and pigs [32]. The most commonly used scales to measure postoperative pain in sheep are still unidimensional, such as the numerical (NS), simple descriptive (SDS), and visual analogue scales (VAS) [33]. However, these instruments exclusively evaluate the intensity of pain, whereas multidimensional or composite scales include sensory, motor and emotional qualities and may be developed to differentiate specific types of pain [29].

To develop the scales a species-specific ethogram is produced to quantify the duration and/or frequency of the behaviours present before and after a painful stimulus. Although previous studies have reported the behavioural descriptors of pain based on an ethogram after a nociceptive stimulus [5,11,20–22,34–40], to our knowledge, there are no validated behavioural scales in the literature to detect acute pain in sheep following solid statistical analysis. The instruments already developed to evaluate acute pain in sheep were based on behavioural changes in lambs submitted to orchiectomy and tail cutting [20] or facial expression in sheep with pododermatitis and mastitis [41]. Another facial scale (sheep grimace) was published after the beginning of our research [42]. To improve the reliability of pain measurement it is necessary to develop an instrument with in-depth statistical validation, as reported in cats [29], cattle [19], horses [24], and pigs [32], by using a blind and random methodology with evidence of validity, reliability, sensitivity, specificity, and a defined analgesic intervention point [43].

The main objective of this study was to validate a behavioural scale to assess acute pain in sheep undergoing soft tissue surgery (laparoscopy). The authors first constructed an ethogram and included pain behaviours described in the literature, then used videos from this study for

further refinement and to define a cut-off point for analgesic intervention. The authors hypothesise that the final scale produced in the current study is reliable and demonstrates content, construct, and criterion validities.

## Material and methods

This was a blind, randomised, prospective and opportunist study. The study was approved by the Ethics Committee on Animal Use from the School of Veterinary Medicine and Animal Science, São Paulo State University (Unesp), Botucatu, São Paulo, Brazil, under protocol 0027/ 2017 and followed the recommendations of ARRIVE [44] adapted to the experimental design. For the pilot study, five sheep separate from the main study were filmed and evaluated before and after laparoscopy, to choose the best anaesthetic protocol and to test the positioning of the digital camera (Gopro Hero5 Black®) as well as other adjustments, to optimise the quality of filming. For the main study, 48 sheep of the breeds Bergamacia (n = 17), Lacaune (n = 18), and Dorper (n = 13) (*Ovis aries* species, dairy line) from the institution were used; 3.5 ± 1.8 (1.5–6) years of age and weighing 58.5 ± 17.3 (34–92) kg. As inclusion criteria, the sheep were considered healthy through clinical and laboratory evaluation (haematocrit, plasma protein, glucose and lactate). During the study period 4 to 5 or 2 to 3 sheep were housed in large (3 x 2 x 1.1m, length x width x height) or small (2.2 x 2 x 1.2m) pens respectively, where they had previously been routinely housed to be protected from rain or low temperatures. The sheep were habituated to the pens for 24 hours before the start of the study, during which they fasted for feed, and for 12 hours they fasted for water. After completion of the study, the animals were maintained for reproduction in a semi-extensive system and were not used for any other research.

Immediately before surgery, 30,000 IU/kg of benzathine penicillin (Pentabiótico®, Zoetis, São Paulo, SP, Brazil) was administered intramuscularly (IM). After dissociative anaesthesia with 0.5 mg/kg of diazepam (Compaz®, Cristália, Itapira, SP, Brazil) and 5 mg/kg of ketamine (Cetamin®, Syntec; Santana de Parnaíba, SP, Brazil) administered intravenously (IV), lumbosacral epidural anaesthesia was performed with 0.1 ml/kg (1 mg/kg) of 1% lidocaine without vasoconstrictor (Xylestesin®, Cristália, Itapira, SP, Brazil) and anaesthetic infiltration with up to 2.5 mg/kg of 1% lidocaine without vasoconstrictor (Xylestesin®, Cristália, Itapira, SP, Brazil) at the incision site and subsequent introduction of a trocar. When the animals demonstrated any sympathetic response related to the surgery, characterised by an increase of more than 20% in heart rate concerning the value observed before the beginning of the surgery or signs of pain characterised by any movement, dissociative anaesthesia was supplemented with 5 mg/kg of ketamine IV.

In all animals, the same experienced surgeon performed a laparoscopy for follicular aspiration and replacement of follicular cells [45], by inserting three trocars (5 mm) in three retroumbilical regions. The postoperative analgesic intervention was performed after the M2 evaluation, with 0.5 mg/kg meloxicam 2% (Maxicam®, Ourofino, Cravinhos, SP, Brazil) and 0.2 mg/kg morphine (Dimorf®, Cristália, Itapira, SP, Brazil) IV in separate syringes.

### Data collection

Two to six animals underwent surgery per day. The procedures started at 9 am and the evaluations of the last animals ended around 7 pm; the 24-hour measurement (M4) occurred the next day. The study was carried out in the months of April and May 2017, with mean daily temperature and humidity of the environment varying between 16–24°C and 68–92%, respectively. The location had the following geographic coordinates: latitude - 22°51' S; longitude - 48°26' O; altitude—818 m.

| Moments Filming | M1 1h before surgery | Anesthesia (Diazepam + Ketamine IV) Local anesthesia: Lidocaine (epidural) + local infiltration | Surgery | M2 3-4h after surgery | Rescue Analgesia (Meloxicam + Morphine IV) | M3 1h after rescue analgesia | M4 24 h after surgery |
|---|---|---|---|---|---|---|---|

**Fig 1. Timeline of moments for validation of the Unesp-Botucatu sheep acute composite pain scale (USAPS).**

Video recordings from 48 sheep were taken at the following moments: M1—one hour before surgery; M2—at the predicted time of greatest pain, between three and four hours after the end of surgery; M3—one hour after the analgesic intervention; and M4–24 hours after surgery (Fig 1).

The in-person observer (NEOFS) made these recordings using a digital camera positioned on a tripod placed outside the pens. The observer turned on the camera and distanced themself at least 10 m from the pens to minimise the effect of human presence on the sheep behaviour. At the end of each recording, the observer approached the sheep and took frontal, lateral and oblique photographs of the sheep's face with a digital camera (Sony Alpha A6500®). These photographs were used to assess the facial pain scale [41].

To analyse the pain-related behaviour in sheep, the research was divided into the following phases: 1) elaboration of an ethogram to characterise the behaviour of the animals before and after the painful procedure (S1 Table); 2) content validation of the normal and pain behaviours based on previous studies, the pilot study, and the ethogram [5,11,20,34–38]. (S2 Table); 3) production of a pre-refinement scale (S3 Table—scale 1), used to evaluate the videos, by four observers blind to the moments; 4) statistical analysis of the pre-refinement scale (S3 Table) evaluated by the observers according to the criteria in Table 1; 5) refinement criteria applied to the scale (S4 Table), based on the statistical analysis of Table 1; 6) validation of the final scale (scale 2) after refinement (Table 3) and presentation of data analysis in the results (Fig 2).

## Ethogram

The observer (NEOFS) watched the 20-minute videos of all moments described in Fig 1 twice (48 animals x 4 moments = 192 videos; a total of 64 hours). The observer watched the videos for the first time for recognition and selection of the relevant pain behaviours. During the second viewing, the observer registered the duration of the behaviours according to the focal animal method [58]. The observer calculated the proportion of each behaviour duration concerning the total recording time of 20 minutes. The ethogram was composed of these

**Table 1. Statistical methods for refinement (R) and validation (V) of the Unesp-Botucatu sheep acute pain composite scale (USAPS).**

| Type of analysis | Description | Statistical test |
|---|---|---|
| **Content validation** R | The following steps were performed: 1) a list of pain-related behaviours reported in the literature and 2) behaviours observed in the ethogram were scored by 3) a committee composed of three veterinarians experienced in assessing pain in ruminants which analysed each subitem within each item of the scale into relevant (+1), do not know (0), or irrelevant (-1). | All the values of each subitem (-1, 0, or 1) were added and the total was divided by the number of observers. Items with a total score $> 0.5$ were included in the scale [46]. |
| **Distribution of scores** V* | Distribution of the frequency of the presence of the scores 0, 1, and 2 of each item at each moment and in all moments grouped (MG). | Descriptive analysis. |

*(Continued)*

**Table 1.** (*Continued*)

| Type of analysis | Description | Statistical test |
|---|---|---|
| **Multiple association** [RV*] | The multiple association of the items with each other was analysed at all moments grouped (MG) using analysis of main components, to define the number of dimensions determined by different variables that establish the scale extension. | **Principal component analysis** ("princomp" and "get_pca_var" functions from the "stats" and "factoextra" packages respectively). According to the **Kaiser** criterion [47], representative dimensions of the components were selected with eigenvalue > 1 and variance > 20 and each item on the scale with a load value ≥ of 0.50 or ≤ - 0.50. For the biplot, confidence ellipses were produced with significant levels of 95% to show the density of scores at each moment. |
| **Intra-observer reliability** [RV] | **Repeatability**—the level of agreement of each observer with themself was estimated by comparing the two phases of assessment, using the scores of each item, the total sum of the USAPS, NS, SDS, VAS, and the need for rescue analgesia. | For the scores of the items of the USAPS and the NS and SDS, and the need for rescue analgesia, the **weighted kappa coefficient** ($k_w$) was used; the disagreements were weighted according to their distance to the square of perfect agreement. The 95% confidence interval (CI) $k_w$ ("cohen.kappa" function of the "psych" package) was estimated. For the VAS, the **intraclass correlation coefficient (ICC)** type "**agreement**" was used and its 95% CI ("icc" function of the "irr" package) [48–50]. For the sum of the USAPS, the **consistency type ICC** and its 95% CI were used. Interpretation of $k_w$ and ICC: very good 0.81–1.0; good 0.61–0.80; moderate 0.41–0.60; reasonable 0.21–0.4 and poor < 0.2. [29,51,52]. The $k_w$ and ICC > 0.50 were used as a criterion to refine the scale. |
| **Inter-observer reliability** [RV*] | **Reproducibility** (**agreement matrix**)—a matrix was generated to assess the level of agreement among all observers, using the scores for each item, the total sum of the USAPS, NS, SDS, VAS, and the need for rescue analgesia. | |
| **Criterion validity** [RV*] | **1) Concurrent criterion validity (relationship with a validated instrument)**—the correlation of the sum of the USAPS was estimated with the NS, SDS, VAS, and facial expression scale of all grouped moments. | **Spearman** rank **correlation** coefficient (rs; "rcorr" function of the "Hmisc" package). Interpretation of the degree of correlation rs (p < 0.05): 0–0.35 low correlation; 0.35–0.7 moderate correlation; 0.7–1.0 high correlation [29]. |
| | **2) Concurrent criterion validity**—the agreement between each observer vs all other observers (reproducibility). | Please see description above for inter-observer reliability. |
| | **3) Predictive criterion validity**—was assessed by the number of sheep that should receive rescue analgesia according to the Youden index (described below) in the moment of greatest pain (M2). | Descriptive analysis. |
| **Construct validity** [RV*] | **Responsiveness: 1) Ethogram**—the proportion of each behaviour duration concerning the total recording time of 20 minutes observed by the in-person researcher at the four moments of the evaluation was compared over time (M1 vs M2 vs M3 vs M4). | The data distribution was assessed by graphs of boxes and histograms ("boxplot" and "histogram" functions of the "graphics" and "lattice" packages, respectively). As data were considered nonparametric, the **Friedman test** (function "friedman.test" of the package "stats") was used for comparisons over time. The p-value was corrected with the **Bonferroni procedure** (function "pairwiseSignTest" of the package "rcompanion") [19,29]. |
| | **2) Scale**—the scores of each item and the total score of the USAPS, NS, SDS, VAS, and the need for rescue analgesia over time were compared. | For the dichotomous variable **need for rescue analgesia logistic regression analysis** ("glm" function of the "stats" package) was applied using the post hoc **Tukey test** ("lsmeans" function of the "lsmeans" package). The model residuals ("residuals" function of the "stats" package) for the **dependent variables** USAPS, NS, SDS and VAS showed Gaussian distribution according to the quantile-quantile and histogram graphs ("qqnorm" and "histogram" functions of the "stats" and "lattice" packages, respectively), thus, **mixed linear models** ("lme" function of the "nlme" package) were applied. The residual distribution was not normal for other **dependent variables** (interaction, activity, locomotion, head position, appetite, and posture), therefore, **generalised mixed linear models** ("glmer" function of the "lme4" package) were applied. In both cases, the **Bonferroni** test was the post hoc test used [19,29]. **Moments, breeding, observers and phases** were included for all models as fixed effects and the individuals were considered a random effect. For these variables, differences were compared applying the post hoc test and, only to USAPS, moments, breeding and observers were used. |
| | Construct validity was determined using the **three hypothesis test**: 1) if the scale really measures pain, the score after surgery (M2) should be higher than the preoperative score (M1 < M2); 2) the score should decrease after analgesia (M2 > M3); 3) and over time (M2 > M4). | |
| **Item-total correlation** [RV*] | The correlation of each item with the total score, excluding the evaluated item, was estimated to analyse homogeneity, the inflationary items and the relevance of each item of the scale. The analysis was performed for all grouped moments (MG). | **Spearman rank correlation coefficient** (r; "rcorr" function of the "Hmisc" package). Interpretation of correlation r: suitable values 0.3–0.7 [52]. Items were accepted if r > 0,3. |
| **Internal consistency** [RV*] | The consistency (interrelation) of the scores of each item on the scale was estimated. The analysis was performed for all grouped moments (MG). | **Cronbach's alpha coefficient** (α; "cronbach" function of the "psy" package) [51] Interpretation: 0.60–0.64, minimally acceptable; 0.65–0.69 acceptable; 0.70–0.74 good; 0.75–0.80 very good; and >0.80 excellent [53]. |

(*Continued*)

**Table 1.** (Continued)

| Type of analysis | Description | Statistical test |
|---|---|---|
| Specificity [RV*] and Sensitivity [RV*] | The scores of the USAPS at **M1** (for **specificity**) and **M2** (for **sensitivity**) were transformed into dichotomous variables (score "0"—the absence of pain expression behaviour for a given item; scores "1" and "2"—the presence of pain) and applied to the equation. | $Sp_{M1} = \frac{TN}{TN+FP}$ |
| | | $Sp$ = specificity. **TN = true negative** [scores that represented painless behaviours (0) at the time when the animals were expected to have no pain, since it was before surgery—M1]. **FP = false positive** [scores that represented pain (1 or 2) in M1]. |
| | | $S_{M2} = \frac{TP}{TP+FN}$ |
| | | $S$ = sensitivity. **TP = true positive** [scores that represented pain expression behaviours (1 or 2) at the time the animals were expected to have pain since it was after surgery–M2]. **FN = false negative** [scores representing painless behaviours (0) at M2]. |
| | | **Interpretation**: excellent 95–100%; good 85–94.9%; moderate 70–84.9%; not specific or not sensitive < 70%. Only items ≥ 70% were included [52]. |
| | | The pain-free (**M1**) and the most intense pain (**M2**) moments were used as the true values and each item of the USAPS as a predictive value to build a receiver operating characteristic curve (**ROC**) (ROC; "roc" function of the "pROC" package). The area under the curve (**AUC**) and its 95% confidence interval (CI) was calculated replicating the original ROC curve 1,001 times by the bootstrap method ("ci.coords and "ci.auc" functions of "pROC" package). |
| Rescue analgesic point [V*] | The **need for analgesia** according to the clinical experience, after the observers had watched the videos, was used as the true value and the **total score of the USAPS** as a predictive value to build a **ROC curve**. The cut-off point for rescue analgesia was determined based on the **Youden index** and its **diagnostic uncertainty zone** using all moments of pain assessment on the USAPS. **Cut-off point** was represented by the Youden index using all moments of pain assessment on the USAPS, NS, SDS and VAS. The **AUC** was calculated and indicates the discriminatory capacity of the test. The **frequency and percentage of animals scored in the diagnostic uncertainty zone** of the cut-off point for the USAPS, NS, SDS, VAS were calculated using descriptive statistical analysis. | $YI = (S + Sp) - 1$ |
| | | $YI$ = **Youden Index**; $S$ = sensitivity; $Sp$ = specificity. Analysis of the **ROC curve** (ROC; "roc" function of the "pROC" package) and the **AUC**: graphical representation of the relationship between the "TP" (S) and the "FP" (1-$Sp$). $YI$ is the point of greatest sensitivity and specificity simultaneously, determined by the ROC curve. **Interpretation:** AUC ≥ 0.95—high discriminatory capacity of the scale [54]. The **diagnostic uncertainty zone** was determined by two methods, calculating: 1) the 95% CI replicating the original ROC curve 1,001 times by the bootstrap method ("ci.coords" and "ci.auc" functions of "pROC" package); 2) the interval between the sensitivity and specificity values of 0.90. The highest interval of one of these two methods was considered the diagnostic uncertainty zone, which indicates the diagnostic **accuracy** [55,56]. |

Scales: numerical (NS), simple descriptive (SDS), visual analogue (VAS).

*The validation analyses were performed using the scores given at all time-points by all evaluators grouped in phases 1 and 2. Statistical analysis was performed using R software in the RStudio integrated development environment [57]. For all analyses, an α of 5% was considered, MG—data of grouped moments (M1 + M2 + M3 + M4)

behaviours (S1 Table) and used to build the pain scale. Next, the videos were edited with the inclusion of the predominant behaviours for a period of about three minutes at each moment. The edited videos were evaluated by four observers for the scale validation process.

## Pain scale video evaluation

The videos were made available to the observers on a virtual platform. The four moments were randomised for each animal and the observers were blind to the moment the videos were recorded. At the end of the observation of each video, the observers, based on their clinical experience, answered whether or not they would administer rescue analgesia (0 = no and 1 = yes to administer rescue analgesia). These data were used to determine the cut-off point related to the need for analgesic intervention. Next, pain scores were determined using three unidimensional scales (NS, SDS, and VAS), the composite pain scale (S3 Table) and the facial scale by observing the photographs included at the end of each video recording [41].

The NS ranges from "0" to "10", where "0" represents no pain and "10" the worst possible or imaginable pain; the SDS ranges from 1—no pain, 2—mild pain, 3—moderate pain, and 4—severe pain; and the VAS is based on a straight line 100 mm long, where "0" represents the animal without pain and "100" the worst possible pain [23,26,27].

## Statistical analysis

Statistical analysis was performed using R software in the RStudio integrated development environment [57]. For all analyses, an α of 5% was considered. Table 1 presents the methods for the refinement and statistical validation process of the proposed scale. The pre-refinement scale used for evaluation of the videos (S3 Table) was submitted to statistical analysis according to Table 1. To produce the validated final scale, the inclusion and exclusion criteria of items and subitems followed ten statistical tests (S4 Table): 1) ethogram (significantly longer duration of each behaviour according to the Friedman test at M2 vs the other moments); 2) content validation (S2 Table); 3) at least 15% frequency of occurrence of each behaviour at M2; 4) multiple association (principal component analysis); 5) intra-observer reliability; 6) inter-observer reliability; 7) construct validity—higher score of the behaviour at M2 vs at least two of the three moments (M1, M3 and M4) according to Friedman test; 8) item-total correlation; 9) internal consistency; 10) specificity and sensitivity. The behaviours that met the criteria stipulated in more than 50% of these statistical tests were accepted and included in the final scale.

The statistical analysis used for the validation of the scale encompassed data from phases 1 and 2 of all observers.

## Results

### Behaviour data (ethogram)

S1 Table contains the behaviours recorded in the ethogram and Table 2 contains the percentage duration of each behaviour. When the moment of greatest pain (M2 –postoperative) was compared to the moment when sheep were supposedly pain-free (M1—basal), the following differences were observed: the duration of "normal interaction", "normal locomotion", and "head above the withers" decreased and the duration of "reduced and absent locomotion" and "arch the back" increased. After the rescue analgesia (M3), compared to M2, the duration of "eat" and "normal interaction" increased, and the duration of "normal and reduced/altered locomotion" and "head below the withers" decreased. "Eat" increased at M3 compared to M2. It was not possible to compare appetite between M2 *vs* M1 because sheep were fasting before surgery.

### Pain scale data

According to the inclusion/exclusion refinement criteria (S4 Table), the following sub-items were excluded: "walks backwards", "walks in a circle", "kicks and stamps limbs on the ground", "extends one or more limbs, "body tremors", and "crawls in ventral recumbence, without getting up."

The final version of the USAPS containing six items (five with three subitems and one with four subitems) was validated (Table 3).

### Distribution of scores

The distribution of scores "0", "1" and "2" occurred as expected, according to the degree of pain. The score "0" predominated at moments M1, M3 and M4. Scores "1" and "2" were more frequent in M2 and decreased in M3. Only the item "activity" of score "1" was not representative. The most frequent postures in M2 were "extends the head and neck" and "lying down with head resting (or close) on (to) the ground" (Fig 3).

**Table 2. Median and range of the percentage duration of behaviours of 48 sheep before and after laparoscopy.**

| Moments | M1 | | M2 | | M3 | | M4 | |
|---|---|---|---|---|---|---|---|---|
| Behaviour category | Median | Range | Median | Range | Median | Range | Median | Range |
| **Eat** | **0[c]** | 0–0 | **11[b]** | 0–82 | **31.25[a]** | 0–86 | 34[ab] | 0–71 |
| Ruminate | 0[b] | 0–24 | 0[ab] | 0–41 | 4.04[a] | 0–20 | 4[a] | 0–44 |
| Drink | 0 | 0–7 | 0 | 0–26 | 0 | 0–16 | 0 | 0–2 |
| Urinate | 0 | 0–9 | 0 | 0–12 | 0 | 0–21 | 0 | 0–5 |
| Defecate | 0 | 0–0 | 0 | 0–19 | 0 | 0–0 | 0 | 0–0 |
| **Normal interaction** | **44[a]** | 0–100 | **0[b]** | 0–91 | **7.15[a]** | 0–95 | **61[a]** | 0–97 |
| **Reduced interaction** | 0[a] | 0–97 | **8[a]** | 0–94 | 0[a] | 0–100 | **0[b]** | 0–97 |
| Absent interaction | 0 | 0–100 | 0 | 0–100 | 0 | 0–30 | 0 | 0–88 |
| **Normal locomotion** | **23[a]** | 0–57 | **0[b]** | 0–68 | **0[c]** | 0–47 | **14[a]** | 0–50 |
| **Reduced/altered locomotion** | **0[b]** | 0–21 | **0[a]** | 0–25 | **0[b]** | 0–40 | 0[a] | 0–47 |
| **Absent/abnormal locomotion** | **0[b]** | 0–70 | **2[a]** | 0–100 | 0[ab] | 0–60 | **0[b]** | 0–30 |
| **Head above the withers** | **11[a]** | 0–80 | **0[b]** | 0–50 | 0[b] | 0–67 | **21[a]** | 0–86 |
| Head at the height of the withers | 0 | 0–68 | 0 | 0–78 | 0 | 0–87 | 0 | 0–81 |
| **Head below the withers** | 10[a] | 0–70 | **13[a]** | 0–81 | **6.46[b]** | 0–60 | **5[b]** | 0–25 |
| Standing still in normal posture | 61[b] | 33–89 | 67[ab] | 0–94 | 73.87[a] | 0–100 | 74[ab] | 10–92 |
| Standing in altered posture | 5[a] | 0–31 | **6[a]** | 0–37 | 1.04[ab] | 0–60 | **0[b]** | 0–8 |
| Kick and stamp the limbs on the ground | 0[b] | 0–0 | 0[ab] | 0–5 | 0[b] | 0–0 | 0[a] | 0–14 |
| Lying down with extension of the head and neck and/or limb(s) | 0[ab] | 0–31 | **0[a]** | 0–38 | 0[ab] | 0–60 | **0[b]** | 0–11 |
| Lying down | 0 | 0–31 | 0 | 0–90 | 0 | 0–35 | 0 | 0–44 |
| Lying down with head turned back | 0 | 0–0 | 0 | 0–0 | 0 | 0–0 | 0 | 0–0 |
| Lying with head supported on or close to the ground | 0 | 0–28 | 0 | 0–100 | 0 | 0–32 | 0 | 0–0 |
| Look at affected area | 0 | 0–0 | 0 | 0–0 | 0 | 0–0 | 0 | 0–0 |
| Lick the affected area | 0 | 0–0 | 0 | 0–0 | 0 | 0–0 | 0 | 0–0 |
| Quick and repeated tail movements | 0 | 0–40 | 0 | 0–0 | 0 | 0–0 | 0 | 0–0 |
| Keep the tail straight | 0 | 0–22 | 0 | 0–6 | 0 | 0–0 | 0 | 0–50 |
| **Arch the back** | **0[b]** | 0–0 | **0[a]** | 0–34 | 0[ab] | 0–13 | 0[ab] | 0–7 |
| Body tremors | 0 | 0–0 | 0 | 0–11 | 0 | 0–0 | 0 | 0–0 |

The proportion of duration of each behavioural category was calculated based on the total time of each period of evaluation (20 mins). Different letters express significant differences between moments (values in bold express differences at M2 compared to M1, or M3, or M4) with a>b>c, according to the Friedman test (p <0.05) [19,29]. M1: preoperative; M2—postoperative, before rescue analgesia; M3—postoperative, after rescue analgesia; M4 - 24h after surgery.

### Principal component analysis

The multiple association among the items of the scale evaluated through principal component analysis selected the main component 1, representative of one dimension, providing the mathematical reason why the scale is unidimensional (Table 4; Fig 4).

### Intra-observer reliability

Repeatability ranged from reasonable to good for each item on the USAPS (except for appetite for evaluator 4 which was poor) and from good to very good for their total score at all moments assessed (Table 5).

### Inter-observer reliability

Inter-observer agreement for all items of the USAPS was moderate for all observers, except for posture and appetite which were reasonable for one observer (S5 Table). The total USAPS

**Table 3. Final validated Unesp-Botucatu sheep acute composite pain scale (USAPS).**

| Item | Subitem (descriptors) | Score | Links to videos |
|------|----------------------|-------|-----------------|
| **Interaction** | Active, attentive to the environment, interacts and/or follows other animals | 0 | https://www.youtube.com/watch?v=4fOJWD-uNbg&t=9s |
| | Apathetic: may remain close to other animals, but interacts little | 1 | https://www.youtube.com/watch?v=EEyMC_VIMpk |
| | Very apathetic: isolated or not interacting with other animals, not interested in the environment | 2 | https://www.youtube.com/watch?v=5NsthhKoEP4 |
| **Locomotion** | Moves about freely, without altered locomotion; when stopped, the pelvic limbs are parallel to the thoracic limbs | 0 | https://www.youtube.com/watch?v=W0Hw2Ibqbyk |
| | Moves about with restriction and/or short steps and/or pauses and/or lameness; when stopped, the thoracic or pelvic limbs may be more open and further back than normal | 1 | https://www.youtube.com/watch?v=i8FxBj-yQhw |
| | Difficulty and/or reluctant to get up and/or not moving and/or walking abnormally and/or limping; may lean against a surface | 2 | https://www.youtube.com/watch?v=dPdT9VMJTi0 |
| **Head Position** | Head above the withers or eating | 0 | https://www.youtube.com/watch?v=W8mi15I1dr8 |
| | Head at the height of the withers | 1 | https://www.youtube.com/watch?v=8xSUmoXaiZY |
| | Head below the withers (except when eating) | 2 | https://www.youtube.com/watch?v=YRxpWSTsqpw |
| **Posture** | Arched back | | https://www.youtube.com/watch?v=gloa-38gTW8 |
| | Extends the head and neck | | https://www.youtube.com/watch?v=rNh_aFePKAE |
| | Lying down with head resting on the ground or close to the ground | | https://www.youtube.com/watch?v=LT6BJzhZO9E |
| | Moves the tail quickly (except when breastfeeding) and repeatedly and/or keeps the tail straight (except to defecate/urinate) | | https://www.youtube.com/watch?v=91RbQMsa8Mg |
| | Absence of these behaviours | 0 | |
| | Presence of one of the related behaviours | 1 | |
| | Presence of two or more of the related behaviours | 2 | |
| **Activity** | Moves normally | 0 | https://www.youtube.com/watch?v=dDx9FesiA2M |
| | Restless, moves more than normal or lies down and gets up frequently | 1 | https://www.youtube.com/watch?v=3MjccV2yV74 |
| | Moves less frequently or only when stimulated using a stick or does not move | 2 | https://www.youtube.com/watch?v=EvLDBJo93jo |
| **Appetite** | Normorexia and/or rumination present | 0 | https://www.youtube.com/watch?v=no1VeiFglUE |
| | Hyporexia | 1 | https://www.youtube.com/watch?v=aIEY1UkqQ-k |
| | Anorexia | 2 | https://www.youtube.com/watch?v=YV40N-OHuNI |

Complete playlist: https://www.youtube.com/watch?v=4fOJWD-uNbg&list=PLTDt73d-ilJNkqldoGmxqMEwc9WzJN0MF

matrix agreement was moderate or good. USAPS was the only scale agreement > 0.50 for all observers (Table 6).

## Criterion validity

**Concurrent criterion validity.** There was a high correlation between USAPS and NS (r = 0.83), SDS (r = 0.81), and VAS (r = 0.81), and moderate correlation with the facial scale (r = 0.48) (Fig 5).

## Construct validity (responsiveness)

The scores for all items and the total score of USAPS were significantly higher at M2 than at M1, M3, and M4, demonstrating their responsiveness. The differences between moments for the total scores of USAPS, the NS, SDS, and VAS were M2 > M3 > M1 > M4 (Table 7; Fig 6).

Evaluators and breeds (as fixed effects) influenced the total score of the USAPS. When pain scores of USAPS were compared separately for breeds, the differences in the total scores of Bergamacia and Lacaune sheep (n = 18) were the same as for all sheep together (M2 > M3 > M1 > M4; Table 7, Fig 6). The differences in the total scores of Dorper sheep were M2 > M3 = M1 > M4. There was no difference in M2 scores between the breeds. Results from

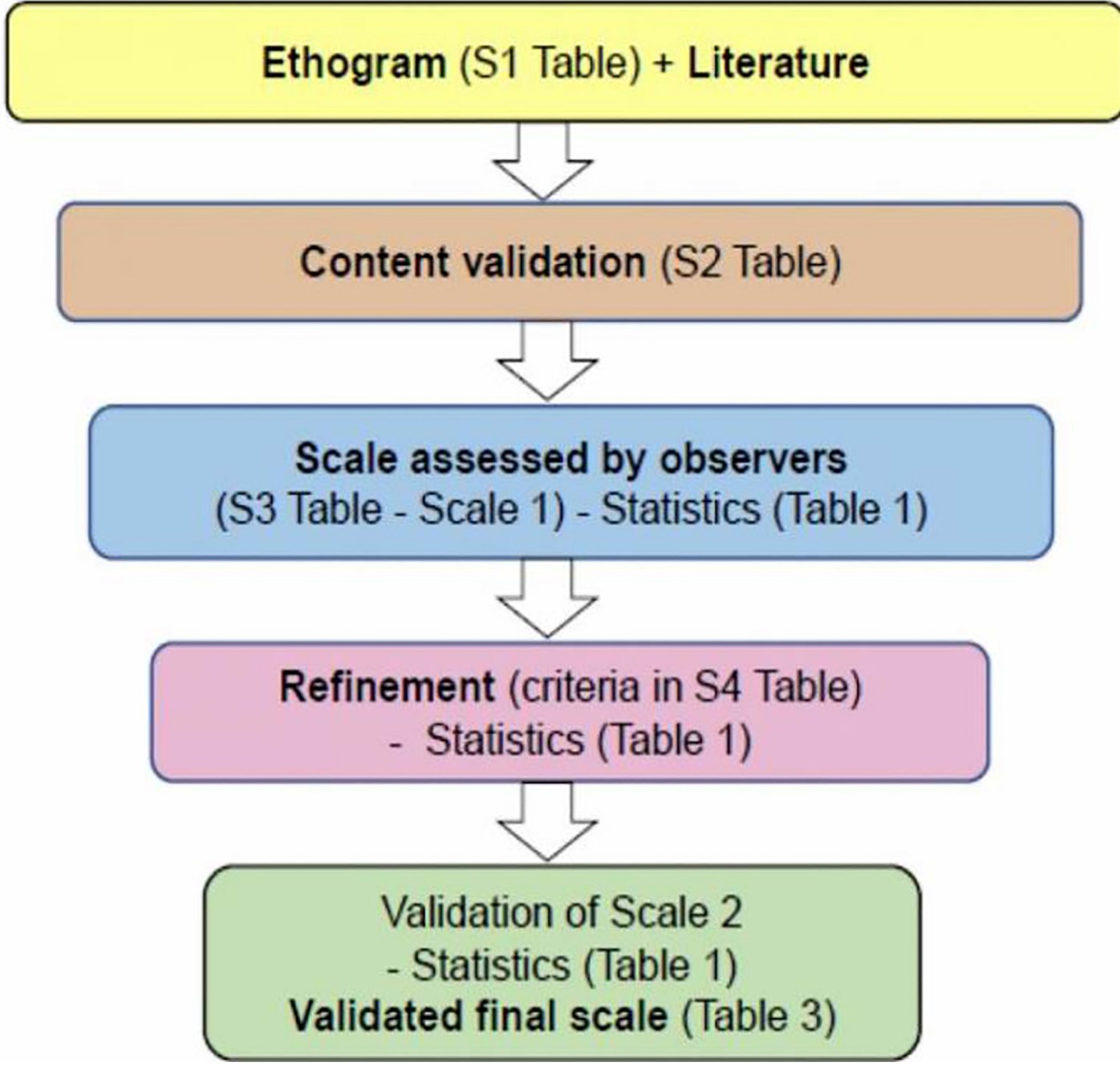

**Fig 2. Flowchart with the stages of elaboration, refinement, and validation of the USAPS.** Statistical tests (Table 1) used in refinement and final validation of the scale: 1) content validation (only in refinement); 2) multiple association; 3) intra-observer reliability; 4) inter-observer reliability; 5) criterion validity; 6) construct validity; 7) internal consistency; 8) sensitivity and specificity; 9) determination of the rescue analgesic point.

two evaluators were the same as for all sheep and evaluators together (M2 > M3 > M1 > M4; Table 7, Fig 6). Results from the other two evaluators were M2 > M3 = M1 > M4, like observed for the Dorper sheep breed. The scores among evaluators were different at M2 [7 (0–12) < 8 (0–12) = 8 (0–12) < 9 (0–12)].

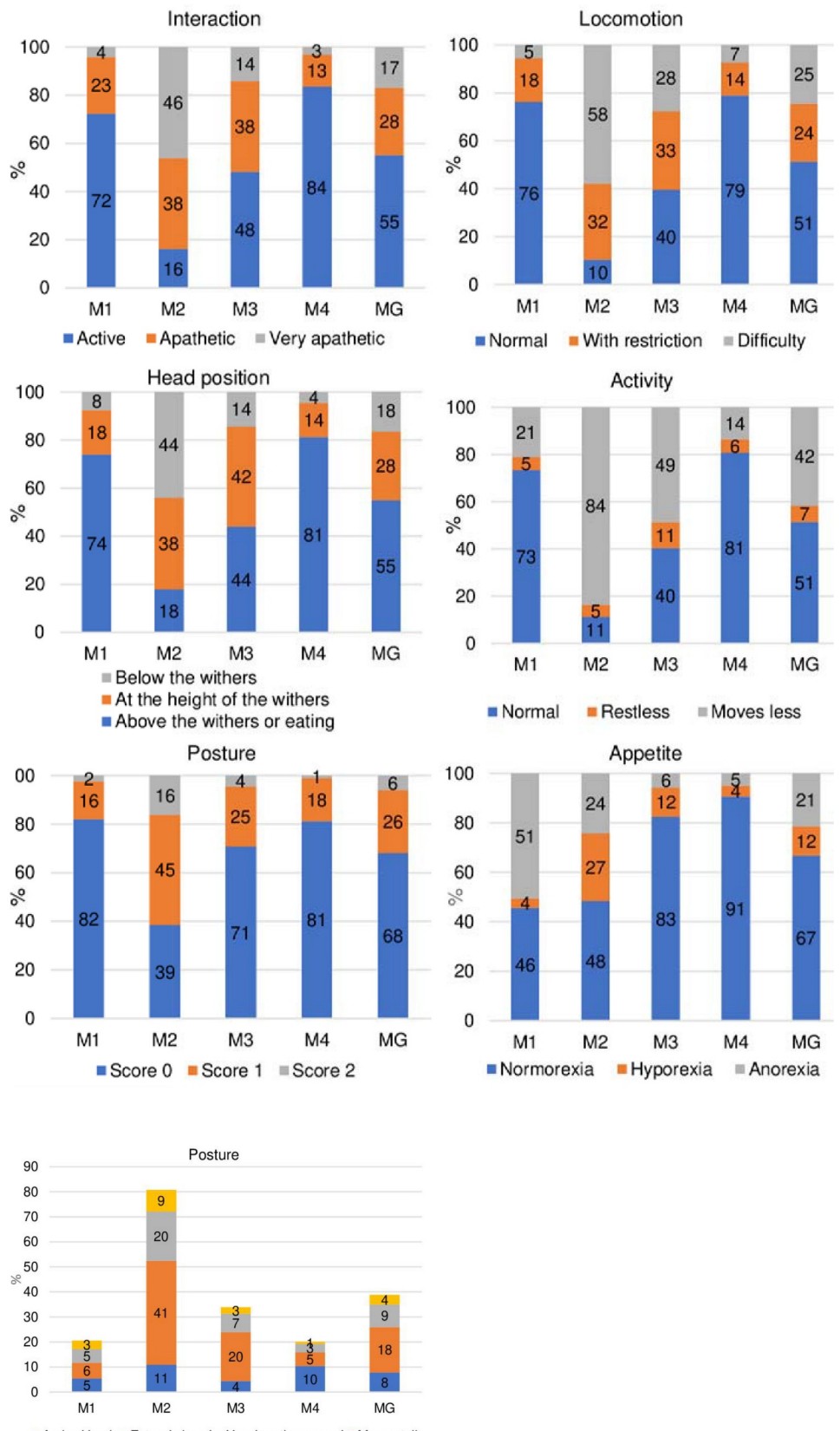

**Fig 3. Frequency of the presence of scores of each item of the USAPS.** Legend: for posture—sum of the scores and individual scores. M1—preoperative; M2—postoperative, before rescue analgesia; M3—postoperative, after rescue analgesia; M4 - 24h postoperative; MG—data of the grouped moments (M1 + M2 + M3 + M4).

## Item-total correlation

The correlation coefficient of item score with the total score (item-total score) ranged from 0.56 to 0.76, therefore all items were accepted, except appetite (Table 8).

## Internal consistency

The Cronbach's α coefficient was 0.81, which indicates that the instrument presents excellent internal consistency and reinforces the possibility of using the full-scale score to interpret the results obtained. Internal consistency was excellent when appetite (0.85) was excluded and very good when all other individual items were excluded, showing that all items contributed similarly and significantly to the total score (Table 8).

## Specificity and sensitivity

All items of the USAPS were specific, except "appetite." All items presented moderate to good sensitivity, except "appetite" and "posture" that were not sensitive (Table 9).

## ROC Curve, Youden index, cut-off point and diagnostic uncertainty zone of the USAPS

In the analysis of the receiver operating characteristic (ROC) curve to determine the cut-off point for diagnosing pain and recommending analgesia, the Youden index was $\geq 4$ of 12 for all grouped evaluators. The interval between the sensitivity and specificity values of 0.90 was between 3.8 and 4. The resampling confidence interval for the Youden index was between 3.5 and 4.5. Based on the resampling result, the diagnostic uncertainty zone scores ranged from 4 to 5; therefore $< 4$ indicates pain-free sheep (true negative) and $> 5$ indicates sheep suffering pain (true positive). The area under the curve was 0.96 (0.95–0.97), indicating that the USAPS presents excellent discriminatory capacity (Fig 7; S6 Table). After exclusion of appetite, the Youden index remained the same ($\geq 4$ of 10) and so did the area under the curve (0.96).

For the unidimensional scales, the cut-off points for rescue analgesia defined by the ROC curve and the Youden index were $\geq 4$ for SN, $\geq 2$ for SDS and $\geq 26$ for VAS (Table 10). Complete data are available in supporting information (S6 Table).

**Table 4. Load values, eigenvalues and variance of the USAPS items based on principal components analysis.**

| Dimensions | 1 | 2 |
|:---:|:---:|:---:|
| **Items** | Load value | Load value |
| Interaction | **0.88** | 0.01 |
| Locomotion | **0.85** | -0.14 |
| Head position | **0.78** | 0.00 |
| Posture | **0.60** | -0.13 |
| Activity | **0.84** | -0.12 |
| Appetite | 0.31 | **0.95** |
| **Eigenvalue** | **3.26** | 0.94 |
| **Variance** | **54.25** | 15.77 |

USAPS–Unesp-Botucatu sheep acute composite pain scale. The structure was determined considering items with a load value $\geq 0.50$ or $\leq -0.50$ (in bold), with representative dimension (eigenvalue > 1 and variance > 20%) [47].

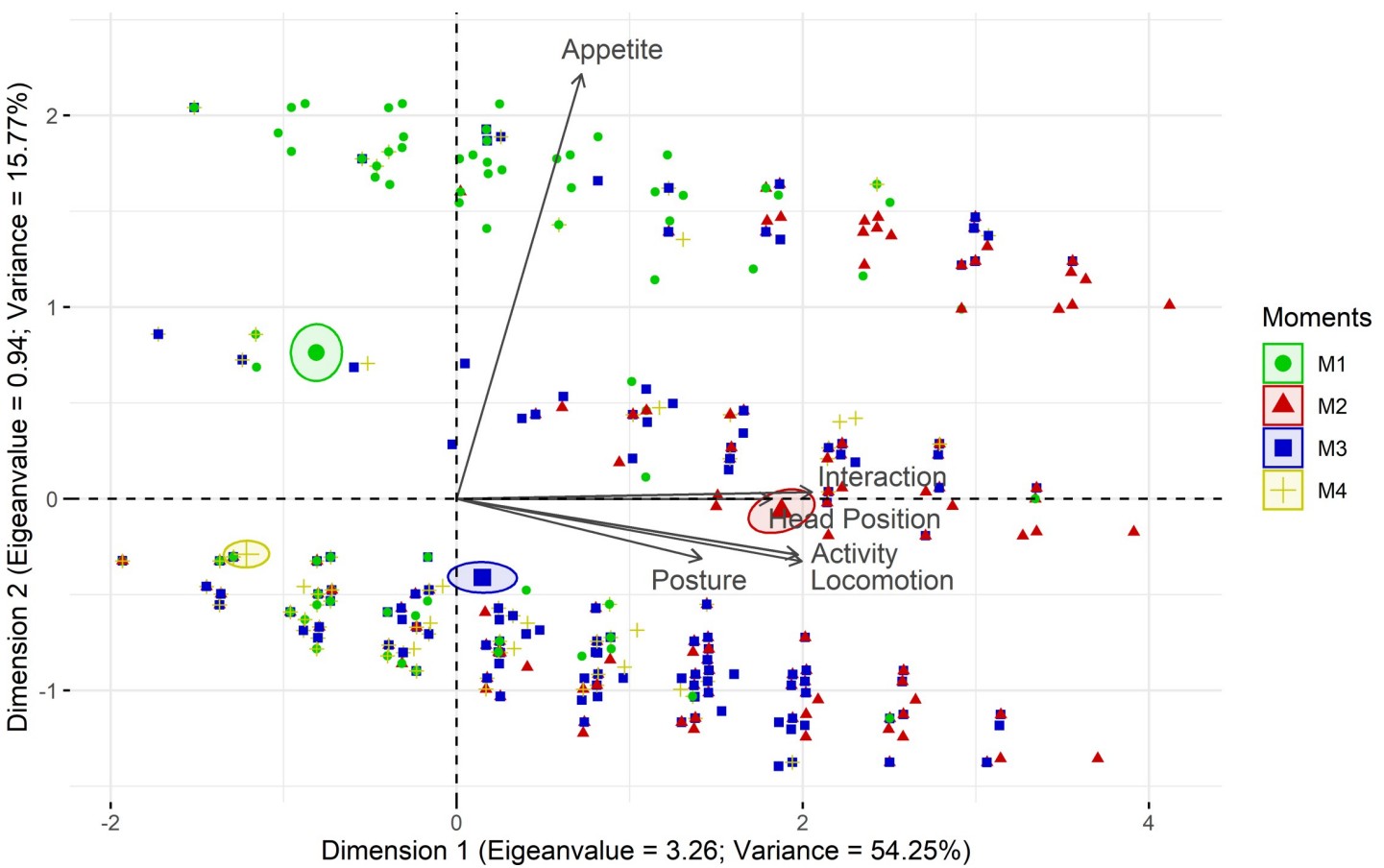

**Fig 4. Biplot of the principal component analysis of the USAPS.** USAPS–Unesp-Botucatu sheep acute composite pain scale. Confidence ellipses were built according to the perioperative moments and pain scores. Moments: M1—preoperative; M2—postoperative, before rescue analgesia; M3—postoperative, after rescue analgesia; M4 - 24h after surgery. Ellipses were constructed according to the moments of pain assessment (M1—green, M2—red, M3 –blue, and M4—yellow). The ellipse referring to the time when sheep were in severe pain (M2) was positioned at the right side of the figure; on the opposite side are the ellipses corresponding to the moments in which sheep were probably not in pain (M1 and M4). The moment of moderate pain (M3) is positioned in the middle. All items on the scale are influenced by moments of pain (M2 and M3) since their vectors are positioned in the direction of these ellipses.

The percentage of animals present in the diagnostic uncertainty zone was low at all moments (13%; 12–15) (Table 11).

**Predictive criterion validity.** Considering the Youden index, 88% (81–96) of sheep would receive rescue analgesia in the moment of most intense pain (M2). Unnecessary analgesia would be indicated in 29% (24–36) of sheep at M1, demonstrating that the scale was sensitive in distinguishing pain and specific in distinguishing sheep not suffering pain (Table 12).

## Discussion

The creation of valid species-specific tools to assess pain is a prerequisite for recognising the phenomenon and determining the need and effectiveness of analgesic treatment. From this perspective, the behavioural pain scale proposed herein is a reliable and valid instrument with a defined analgesic intervention point, which can be used to assess postoperative abdominal pain in sheep. This instrument demonstrates potential clinical applicability to guide decision making for analgesia indication when necessary, and potential experimental applicability for translational studies and those comparing the analgesic efficacy of drugs [1].

**Table 5. Intra-observer reliability of the USAPS, unidimensional scales and rescue analgesia indication in sheep.**

| Evaluator | 1 | | | 2 | | | 3 | | | 4 | | |
|---|---|---|---|---|---|---|---|---|---|---|---|---|
| Items | $k_w$ | Min | Max | $k_w$ | Min | Max | $k_w$ | Min | Max | $k_w$ | Min | Max |
| Interaction | **0.64** | 0.55 | 0.74 | **0.66** | 0.63 | 0.70 | **0.52** | 0.41 | 0.64 | **0.65** | 0.56 | 0.74 |
| Locomotion | **0.71** | 0.64 | 0.79 | **0.65** | 0.59 | 0.73 | 0.48 | 0.36 | 0.60 | **0.61** | 0.52 | 0.71 |
| Head position | **0.67** | 0.57 | 0.76 | **0.71** | 0.67 | 0.74 | 0.48 | 0.36 | 0.60 | **0.59** | 0.49 | 0.68 |
| Posture | 0.41 | 0.22 | 0.60 | **0.67** | 0.67 | 0.67 | 0.48 | 0.34 | 0.62 | **0.58** | 0.55 | 0.61 |
| Activity | **0.54** | 0.44 | 0.64 | **0.65** | 0.58 | 0.71 | **0.56** | 0.47 | 0.66 | 0.47 | 0.35 | 0.59 |
| Appetite | **0.55** | 0.44 | 0.66 | **0.61** | 0.52 | 0.69 | 0.38 | -0.08 | 0.84 | 0.15 | -0.04 | 0.35 |
| Rescue analgesia | **0.67** | 0.56 | 0.77 | **0.75** | 0.65 | 0.85 | **0.53** | 0.41 | 0.65 | **0.53** | 0.40 | 0.65 |
| Numerical scale | **0.80** | 0.80 | 0.80 | **0.85** | 0.85 | 0.85 | **0.58** | -0.19 | 1 | **0.72** | 0.72 | 0.72 |
| SDS | **0.78** | 0.78 | 0.78 | **0.77** | 0.77 | 0.77 | **0.61** | 0.61 | 0.61 | **0.67** | 0.61 | 0.74 |
| Scales | ICC | CI | | ICC | CI | | ICC | CI | | ICC | CI | |
| USAPS | **0.77** | 0.71 | 0.82 | **0.84** | 0.79 | 0.88 | **0.65** | 0.56 | 0.72 | **0.72** | 0.64 | 0.78 |
| VAS | **0.80** | 0.74 | 0.85 | **0.81** | 0.76 | 0.86 | **0.55** | 0.44 | 0.64 | **0.71** | 0.63 | 0.77 |

USAPS—Unesp-Botucatu sheep acute composite pain scale; SDS—simple descriptive scale; VAS—visual analogue scale. $k_w$–weighted kappa coefficient; ICC—intraclass correlation coefficient; CI—Confidence interval. Interpretation of reliability—very good 0.81–1.0; good 0.61–0.80; moderate 0.41–0.60; reasonable 0.21–0.4; poor <0.2 [29,51,52]. Bold type corresponds to values > 0.50.

The validation process of an instrument to assess pain is based on the investigation of behaviours and, when possible, of species-specific physiological data present during pain situations, followed by a comparison of these changes with the state of normality [19,29,32]. This methodology was followed in the current study; an ethogram was constructed during the preoperative period when animals were supposedly devoid of pain, followed by the postoperative period when animals probably had severe pain, followed by rescue analgesia for pain reduction and reassessment after 24 hours. Thus, the experimental design tested the instrument at different pain intensities. The ethogram, together with the pain expression behaviours in sheep described in the literature, served as a basis for the construction of the scale. After content analysis, the first instrument was defined to include relevant behaviours and exclude irrelevant behaviours, to make the instrument as simple and representative as possible.

Filming using video cameras adds value to the data as it enables to archive of the material for future research and minimises the influence of the observer in the evaluation, avoiding possible observer-related behavioural changes that the animal may present which are inherent to

**Table 6. Inter-observer matrix agreement of the USAPS, unidimensional scales and rescue analgesia indication.**

| Evaluator | 1 | 2 | 3 | 1 | 2 | 3 | 1 | 2 | 3 |
|---|---|---|---|---|---|---|---|---|---|
| Scales—$k_w$ | Numerical scale | | | Simple descriptive scale | | | Rescue analgesia | | |
| 2 | 0.40 | | | **0.68** | | | **0.51** | | |
| 3 | 0.44 | **0.69** | | **0.52** | **0.59** | | 0.49 | **0.65** | |
| 4 | **0.72** | 0.46 | 0.49 | **0.67** | **0.65** | 0.47 | **0.55** | 0.47 | 0.43 |
| Scales—ICC | USAPS | | | VAS | | | | | |
| 2 | **0.65** | | | 0.24 | | | | | |
| 3 | **0.57** | **0.74** | | 0.33 | **0.64** | | | | |
| 4 | **0.70** | **0.63** | **0.53** | **0.70** | 0.40 | 0.46 | | | |

USAPS—Unesp-Botucatu sheep acute composite pain scale; VAS—visual analogue. $k_w$–weighted kappa coefficient; ICC—intraclass correlation coefficient; Interpretation of reliability: very good 0.81–1.0; good 0.61–0.80; moderate 0.41–0.60; reasonable 0.21–0.4; poor < 0.2 [29,51,52]. Bold type corresponds to values > 0.50.

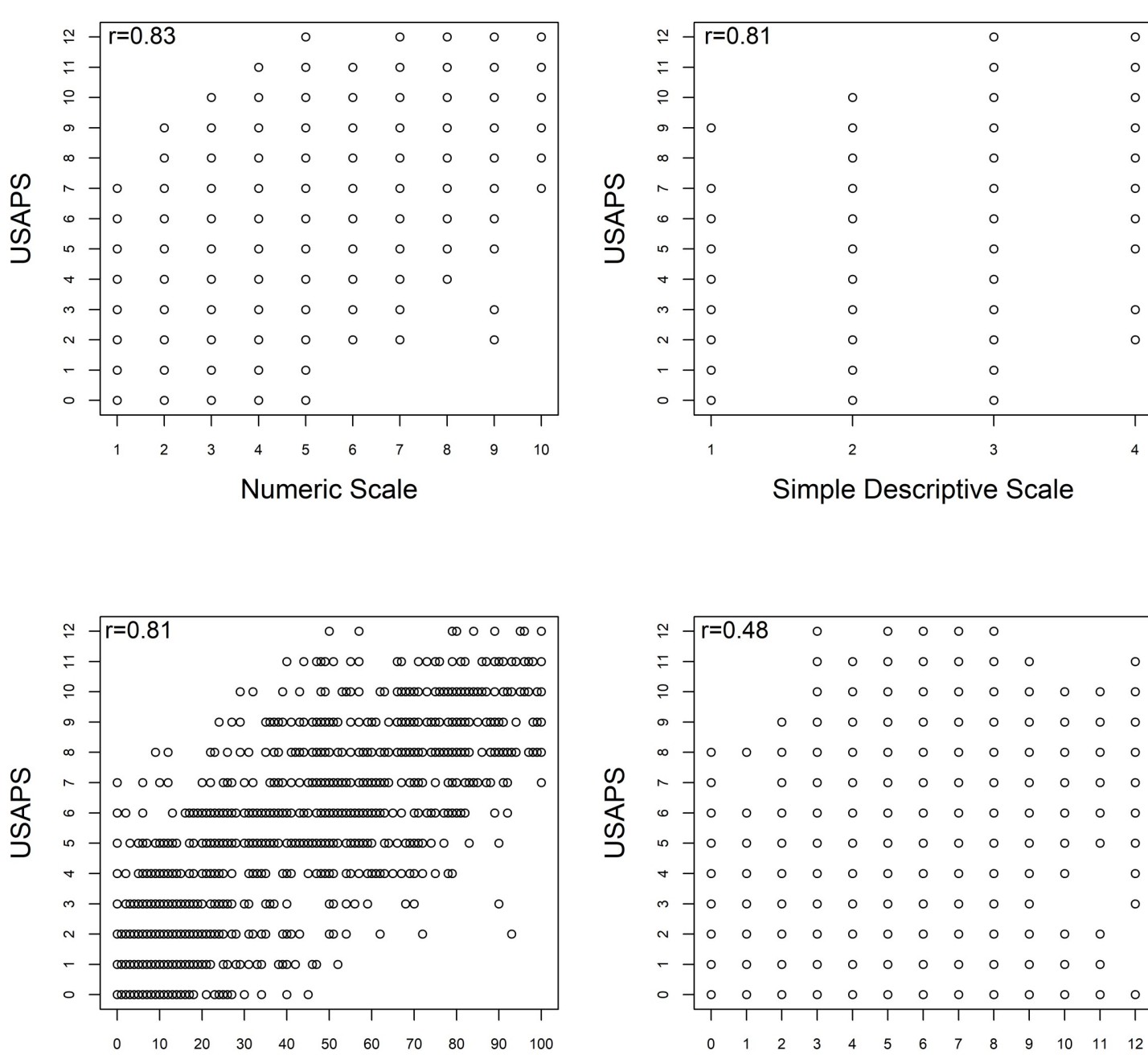

**Fig 5. Spearman correlation between the scores of the USAPS (Unesp-Botucatu sheep acute composite pain scale) and those of the numerical, simple descriptive, visual analogue and facial expression scales.**

the presence of the observer. On the other hand, it is often not possible in a clinical situation to make a remote assessment, and the presence of the observer could interfere with the animals' behaviour [59,60]. Thus, it is still necessary to validate the proposed scale in clinical situations and with the presence of the observer, to ensure that these results are reproducible.

According to the ethogram sheep in pain ate less than after rescue analgesia, which justified the introduction of appetite as one of the criteria evaluated on the scale. At the preoperative

**Table 7. Responsiveness of the USAPS, rescue analgesia and unidimensional pain scales, between the four perioperative moments.**

| Scales | Moments | | | | | | | |
|---|---|---|---|---|---|---|---|---|
| | M1 | | M2 | | M3 | | M4 | |
| Items | Median | Amplitude | Median | Amplitude | Median | Amplitude | Median | Amplitude |
| Interaction | 0[c] | 0–2 | 1[a] | 0–2 | 1[b] | 0–2 | 0[d] | 0–2 |
| Locomotion | 0[c] | 0–2 | 2[a] | 0–2 | 1[b] | 0–2 | 0[c] | 0–2 |
| Head position | 0[c] | 0–2 | 1[a] | 0–2 | 1[b] | 0–2 | 0[d] | 0–2 |
| Posture | 0[c] | 0–2 | 1[a] | 0–2 | 0[b] | 0–2 | 0[c] | 0–2 |
| Activity | 0[c] | 0–2 | 2[a] | 0–2 | 1[b] | 0–2 | 0[d] | 0–2 |
| Appetite | 2[c] | 0–2 | 1[a] | 0–2 | 0[b] | 0–2 | 0[d] | 0–2 |
| **USAPS** | **2[c]** | **0–11** | **8[a]** | **0–12** | **4[b]** | **0–11** | **0[d]** | **0–10** |
| **Rescue analgesia** | 0[c] | 0–1 | 1[a] | 0–1 | 1[b] | 0–1 | 0[c] | 0–1 |
| **NS** | 2[c] | 1–8 | 6[a] | 1–10 | 3,5[b] | 1–10 | 1[c] | 1–9 |
| **SDS** | 1[c] | 1–3 | 3[a] | 1–4 | 2[b] | 1–4 | 1[c] | 1–4 |
| **VAS** | 10[c] | 0–80 | 58[a] | 0–100 | 28[b] | 0–100 | 6[c] | 0–91 |

USAPS—Unesp-Botucatu sheep acute composite pain scale; RA—Rescue analgesia (0—no; 1—yes); NS (1–10), SDS (1–4) and VAS (0–100). Different letters express significant differences between moments where a > b > c > d, according to the mixed linear model [19,29]. M1: preoperative; M2: postoperative, before rescue analgesia; M3: postoperative, after rescue analgesia; M4: 24h postoperative.

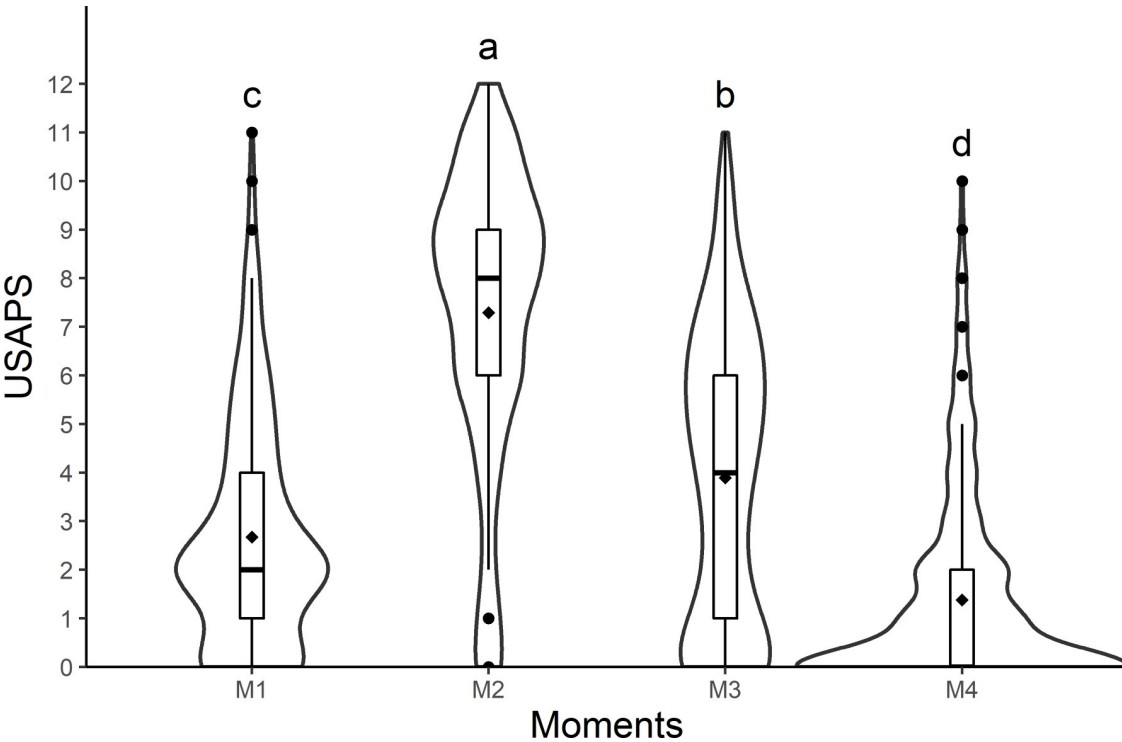

**Fig 6. Violin plot of the scores (median/amplitude) of the USAPS, comparing the four perioperative moments in sheep submitted to abdominal surgery.** The top and bottom box lines represent the interquartile range (25 to 75%), the line within the box represents the median, the extremes of the whiskers represent the minimum and maximum values, black lozenge (♦) represent the mean, black circles (•) represent outliers and width of the figures represent the distribution of data (wider sections represent a larger number of data). USAPS: Unesp-Botucatu sheep acute composite pain scale. Different letters express significant differences between moments where a > b > c > d, according to the mixed linear model [19,29]. M1—preoperative; M2—postoperative, before rescue analgesia; M3—postoperative, after rescue analgesia and M4 - 24h postoperative.

**Table 8. Item-total correlation and internal consistency of the USAPS.**

| Items \ Tests | Item-total (Spearman) | Internal consistency (Cronbach's α) |
|---|---|---|
| Full scale | | 0.81 |
| Excluding each item below | | |
| Interaction | 0.76 | 0.73 |
| Locomotion | 0.72 | 0.74 |
| Head position | 0.62 | 0.77 |
| Posture | 0.56 | 0.80 |
| Activity | 0.71 | 0.75 |
| Appetite | 0.25 | 0.85 |

USAPS: Unesp-Botucatu sheep acute composite pain scale. Interpretation of Spearman's rank correlation coefficient (*r*): 0.3–0.7: acceptable values in bold [52]. Interpretation of the Cronbach's α coefficient values: 0.60–0.64 minimally acceptable; 0.65–0.69 acceptable; 0.70–0.74 good; 0.75–0.80 very good; > 0.80 excellent [53]; bold values > 0.70.

moment, this behaviour was not present because animals were fasting. Decreased appetite is a common finding in sheep submitted to castration and laparoscopy [20,38,45,61,62].

In the current study, laparoscopy led to behavioural alterations indicative of acute pain. Sheep reduced their locomotion and interaction with the environment, lowered their head, and arched their back. Some of these behaviours only returned to normal 24 hours after surgery. These behaviours were similar to those previously reported in lambs undergoing different painful stimuli [21], like mulesing [63] and orchiectomy [64,65]. During severe pain limb, tail, and head movements and full extension of the pelvic limbs occur; during moderate pain vocalisation, standing, sitting, and lying positions with the partial extension of the pelvic limbs or tremor are observed; and during mild pain or no pain postures may be normal [20].

Vocalisation could be a possible indicator of pain, like in other ruminants such as cattle [19,66–68] and goats [68]. However, in sheep, this behaviour is more related to social isolation and restraint. Except at times when the feed was supplied, vocalisation was not observed in the current study and is not an indicator of postoperative pain in adult sheep submitted to laparoscopy, as reported previously in lambs undergoing castration and tail docking [20,69]. Different from the current study, the majority of studies that evaluated acute pain in sheep used lambs [5,20–22,61,63–65], which could limit the extrapolation of the results to adult animals. Some behaviours more specific to lambs mentioned in the literature, such as "jumping like a rabbit", did not occur, as they are more frequent in young animals up to about 5 months of age than in adult animals. Common behaviour in cattle [19] and cited in lambs [20,61] "look at the flank and lick the painful area" was not observed in the adult sheep in the current work.

**Table 9. Specificity and sensitivity of the USAPS.**

| Items \ Tests | Specificity (%) | Sensitivity (%) | AUC | Min. | Max. |
|---|---|---|---|---|---|
| Interaction | 72 | 84 | 0.78 | 0.75 | 0.81 |
| Locomotion | 76 | 90 | 0.83 | 0.80 | 0.86 |
| Head position | 74 | 82 | 0.78 | 0.75 | 0.81 |
| Posture | 82 | 61 | 0.72 | 0.69 | 0.75 |
| Activity | 73 | 89 | 0.81 | 0.78 | 0.84 |
| Appetite | 46 | 52 | 0.49 | 0.45 | 0.52 |

Interpretation of specificity and sensitivity: excellent 95–100%; good 85–94.9%; moderate 70–84.9%; not specific or sensitive <70%; bold values ≥ 70% [52].

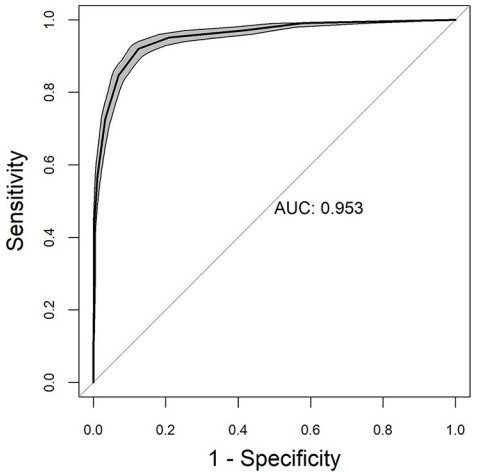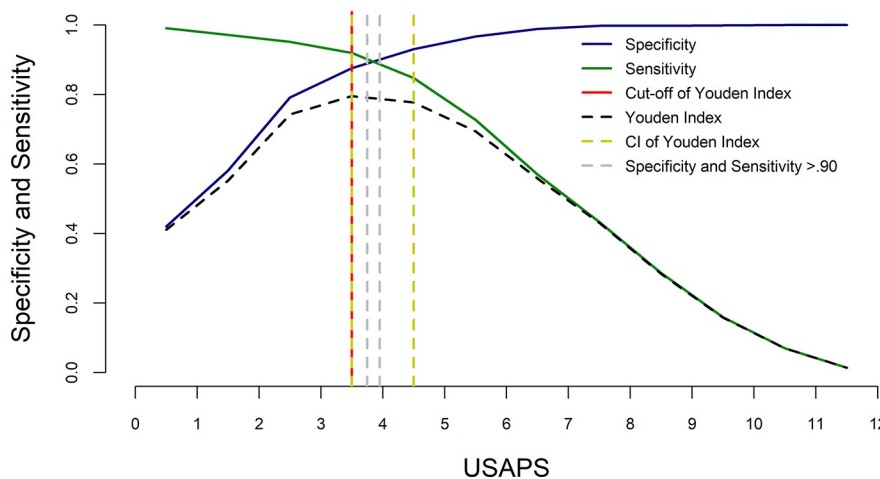

**Fig 7. ROC curve and AUC and two-graph ROC curve with the diagnostic uncertainty zone for the USAPS.** ROC (receiver operating characteristic) curve with a 95% confidence interval (CI) calculated from 1,001 replications and area under the curve (AUC)[54]. Interpretation of AUC $\geq$ 0.95—high discriminatory capacity. Two-graph ROC curve, CI of 1,001 replications, and of sensitivity and specificity > 0.90 applied to estimate the diagnostic uncertainty zone of the cut-off point of all grouped evaluators, according to the Youden index for the Unesp-Botucatu sheep acute composite pain scale (USAPS) [55,56]. The diagnostic uncertainty zone was 4 to 5; < 4 indicates pain-free sheep (true negative) and > 5 indicates sheep suffering pain (true positive). The Youden index was $\geq$ 4, which is representative of the cut-off point for the indication of rescue analgesia.

Validity and reliability are essential attributes for an instrument to identify and quantify pain in animals. The instrument was subjected to a blinded and random methodology, through the same experimental design and recognised scientific robustness [43] already used in cats [29], cattle [19], horses [24], and pigs [32]. Observers familiar with the behaviour of ruminants validated the content of the scale by assessing the representativeness of each item. This analysis measures the extent to which the instrument reflects the phenomenon of interest, in this case, pain [29,30,70]. The evaluators were selected for their experience to improve the reliability and accuracy of the tool based on repeatability and reproducibility [70].

The initially proposed scale contained 33 variables including items, subitems and sub-divisions (S3 Table). Some of these were excluded according to the criteria of the statistical tests (Table 1). The scale refinement identified the 12 most relevant and appropriate items and sub-items to measure pain in sheep and was essential to improve the quality of the validated final scale into a simpler and more objective version [71].

The analysis of score distribution provides an overview of the occurrence of each score at each moment to indicate the importance of each score. The results were as expected since the score 0 (absence of pain) prevailed before and 24h after the surgery, scores 1 and 2 occurred more in the postoperative period and after the rescue analgesia, and the score 2 was more present after surgery, suggesting a greater intensity of pain. The results of each item generally

**Table 10. Scores, specificity, sensitivity and Youden index corresponding to rescue analgesia indication of the USAPS and unidimensional scales.**

| Scale | Score | Specificity | Sensitivity | Youden index |
|:---:|:---:|:---:|:---:|:---:|
| USAPS | 4 | 0.88 | 0.92 | 0.80 |
| NS | 4 | 0.97 | 0.93 | 0.90 |
| SDS | 2 | 0.84 | 0.99 | 0.83 |
| VAS | 26 | 0.94 | 0.94 | 0.88 |

Scales: USAPS—Unesp-Botucatu sheep acute composite pain scale; NS–numerical; SDS—simple descriptive; VAS—visual analogue.

**Table 11. Percentage of sheep present in the diagnostic uncertainty zone according to the Youden index of the USAPS.**

| Moments | Evaluator | | | | |
|---|---|---|---|---|---|
| | 1 | 2 | 3 | 4 | All |
| M1 | 13 | 17 | 15 | 18 | 15 |
| M2 | 13 | 13 | 9 | 3 | 9 |
| M3 | 22 | 19 | 20 | 24 | 21 |
| M4 | 15 | 5 | 5 | 7 | 8 |
| MG | 15 | 13 | 12 | 13 | 13 |

Calculation based on 48 sheep evaluated twice by four evaluators. USAPS: Unesp-Botucatu sheep acute composite pain scale. M1—preoperative; M2—postoperative, before rescue analgesia; M3—postoperative, after rescue analgesia; M4 - 24h postoperative; MG—data of grouped moments (M1 + M2 + M3 + M4). The diagnostic uncertainty zone was 3.5–4.5; < 4 indicates pain-free sheep (true negative) and > 5 indicates sheep suffering pain (true positive).

followed the results of the sum of the scale. Only score 1 of the "activity" item was not so evident, showing that it is rare for sheep to move about more than normal or to lie down and get up frequently, as occurs in other species [19,24,29,32]. A considerable percentage of the score 0 (normorexia and/or ruminating) when sheep were fasting at M1 may be a confounding error. The bedding of the stalls was rice straw and the fact that sheep searched for food on the ground possibly provided the impression they were eating and/or ruminating.

The principal component analysis relates the variables of the tool in a grouped manner and calculates the number of dimensions determined by different variables [29] to establish the extension or dimensionality of the scale [72]. These variables are related so that the items that define specific parts of the construct are grouped by multiple association [33]. The Kaiser criterion selected one component, therefore the scale is unidimensional [52], like in cattle [19] and pigs [32]. An instrument is multidimensional, like in cats [25,29], when in addition to pain intensity, it includes qualitative and temporal characteristics, such as sensory, motor, emotional, and cognitive dimensions, which have a high correlation in the experience of pain [29,33,72]. In a validation of the acute pain scale in lambs, the principal component analysis generated two principal components [20]. Unidimensional scales are not as satisfactory as those with more than one dimension, as they only assess the intensity of pain. However, because they are simple, they are easily applicable. In the current study, it is premature to conclude about the number of dimensions of the proposed pain scale in mathematical terms, since only one statistical model was evaluated. The scale includes several biological aspects of pain, such as physiological (appetite), sensory or motor (posture, activity), emotional (interaction

**Table 12. Percentage of sheep rescue analgesia was indicated according to clinical experience and to the Youden index of the USAPS.**

| Moments | Evaluator | | | | | | | | | |
|---|---|---|---|---|---|---|---|---|---|---|
| | 1 | | 2 | | 3 | | 4 | | All | |
| RA | Exp | YI | Exp | YI | Exp | YI | Exp | YI | Exp | YI |
| M1 | 16 | 29 | 10 | 25 | 21 | 24 | 25 | 36 | 18 | 29 |
| M2 | 92 | 92 | 80 | 82 | 79 | 81 | 97 | 96 | 87 | 88 |

Calculation based on 48 sheep evaluated twice for all evaluators (96 assessments). RA–indication of rescue analgesia according to clinical experience scored at the end of each video analysis (Exp) and according to the Youden index of the USAPS (score ≥ 4). USAPS—UNESP-Botucatu sheep composite acute pain scale. M1—preoperative; M2—postoperative, before rescue analgesia; Youden index ≥ 4 is representative of the cut-off point for the indication of rescue analgesia (see Table 10 for results of Youden index).

with other animals and attention to the environment), and temporal (response to analgesia) [29]; therefore, in biological terms, the USAPS is multidimensional. Future studies addressing different pain models, such as orthopaedic, may or may not confirm if this scale is applicable for other types of pain.

The intra- and inter-observer reliability for each item and the total score of the sheep scale was similar to that of cattle [19] and pigs [32], lower than in cats [29] and higher than in horses [24]. When compared with other instruments developed in the sheep species, the scale proposed here presented reliability similar to the scale reported in lambs subjected to acute pain [20]. Compared to a sheep locomotion scale (with a score ranging from 0 to 6) that demonstrated very good intra (91%) and inter-observer (93%) reliability [73], the proposed instrument showed lower results, in which the item "locomotion" presented good intra-observer reliability for most observers and only moderate inter-observer reliability. Another study, in which 10 veterinarians and 10 sheep farmers scored a locomotion scale, obtained very good and good values for intra- and inter-observer reliability, however, the reliability for individual locomotion scores varied from reasonable to moderate [74].

Validity indicates that the instrument can accurately measure what is proposed. There are three types of validity: content analysis, described before, criterium and construct. Criterion validity assesses the measuring efficiency of a scale and includes concurrent and predictive criterion validity. Concurrent validity compares the instrument to existing validated scales [19,24,29,32], by evaluating the instrument and the criterion simultaneously and predictive validity evaluates the criterion after the test. Both methods were used in this study [75]. Every new instrument needs to be compared with another already established and validated tool [43]. For this context, previous instruments were developed for pain assessment in sheep based on behavioural body changes [20] and facial expression [41,42]. The former instrument was not used for comparison because some behaviours were common to our study and correlation would be inflated. Therefore, the facial expression scale was used as a previously validated gold standard model for testing concurrent criterion validity. A second method compared the proposed instrument with the unidimensional scales, following the same criteria applied in cats [29], cattle [19], horses [24] and pigs [32]. The USAPS showed a high correlation with the unidimensional scales, as previously reported for validated scales in other species [19,24,29,32] and a claudication scale in sheep [76]. Otherwise, correlation with the facial scale was only moderate possibly because other breeds of sheep were used in this study and their facial morphology was different from the original study [41].

Construct validity reflects the responsiveness of the scale and examines whether the instrument detects predictable differences between groups or moments [33]. The method tests the hypothesis that time and surgical and analgesic intervention should alter pain scores [29] and has been used to validate scales in veterinary medicine [19,24,29,32]. In this study, the differences observed in the pain scores between the moments, and especially at the expected moment of greatest pain compared to the other moments, confirm that the proposed scale is responsive both to identify intense degrees of pain, as well as moderate degrees, which occurred after rescue analgesia. In cattle, the alterations between scores (M2>M4 = M3 = M1) [19] were slightly different from sheep, where the pain at M4 decreased after M3 and was even lower than M1 (M2>M3>M1>M4). The increase and decrease in pain scores after surgery and rescue analgesia, respectively, also occurred in cats [29], horses [24] and pigs [32], however, differently from sheep, in these species pain scores tended to increase after 24hs. This is possibly related to the different surgical, anaesthetic and analgesic protocols among these species or because the USAPS responds differently. The USAPS scores were lower at 24h postoperatively than the preoperative scores because sheep did not have access to food before surgery, therefore appetite was scored as anorexia for most sheep at this moment.

Although the evaluators and the breeds, unlike the phases, influenced the total score of the USAPS, the differences in results among breeds and evaluators were observed only between the preoperative and 24h postoperative time points. For some breeds and evaluators these scores were not different and for others, the scores at 24h were lower than the preoperative scores. This does not appear to have relevance, showing that, apparently, the scale worked well regardless of the breeds.

Except for appetite, all items of the proposed scale presented an acceptable item-total correlation, as in pigs [32], which demonstrates their relevance and ensures the homogeneity of the tool. The internal consistency of the proposed scale was excellent and similar to cats—0.86 [29], cattle—0.87 [19] and pigs—0.82 [32], which ensures that the scores of the scale items can be added and the total score will be representative of the pain intensity [29]. The similarity of the values when excluding each item demonstrates that they have a similar tendency and importance [53]. The scale was specific for five of six items and sensitive for four. Postural changes were not as frequent compared to the other items.

The analysis of the ROC curve [54] estimated the cut-off point for analgesic intervention in sheep as in previously validated pain scales in cats [29], cattle [19] and pigs [32]. The determination of scores indicative of the need to use analgesics helps professionals in clinical decisions, confirms or not the efficacy of analgesic treatment [29], and may be used to define welfare in animals. The cut-off point was $\geq 4$ and the diagnostic uncertainty zone of all evaluators ensures that sheep with a score of $> 5$ of a total of 12 points are really in pain, while those with a score $< 4$ do not have pain. The low percentage of animals within the zone of diagnostic uncertainty ensures good reliability in making decisions about the indication for rescue analgesia in animals that present pain and, therefore, should receive analgesia. Thus, the proposed scale presents excellent diagnostic accuracy. Although the definition of the score referring to the analgesic intervention point is a good tool, it is emphasised that even if the scores are $< 4$, in some cases additional analgesia may be necessary according to the clinical evaluation, at the discretion of the observer. The cut-off point was $> 4/10$ in cattle [19], $\geq 6/18$ in pigs [32], and for the subscale "expression of pain" in cats it was $> 2/12$ [29].

Appetite was not approved for most of the validation criteria used in this study, and the reason why it was arbitrarily maintained in the scale was based on the fact that lack of appetite is widely described as a sign of pain in sheep [20,45,62], and other species [19,24,29,32] and it is the only physiological variable of USAPS, which could contribute to its biological multidimensionality. Considering that the Youden index was the same without using the appetite data, one can choose either to use or not this information according to each circumstance, without interfering in decision making concerning rescue analgesia.

Previous studies assessed pain scales in sheep submitted to laparoscopy. In one study the mean pain score was 0.3 of 9 [45] and in another study a pain scale ranging from 0 to 6 based on decreased appetite, limited mobility, and back arching, was insensitive, with 90% of animals with a "0" score and 10% with "1" [62]. A recent empirical study on pain in sheep after cardiac surgery was scored; for scores of 0-2/25 there was no intervention, 3-9/25 rescue analgesia was performed, and $\geq 10/25$ multimodal analgesia was performed [77].

In line with the low percentage of animals within the diagnostic uncertainty zone, the high areas under the curve observed in this study ($\geq 0.95$) indicate that the scale has high discriminatory capacity [55]; it correctly classifies individuals with or without pain, results that resembled cattle [19], pigs [32] and the subscale "expression of pain" in cats [29]. The predictive criterion validity was confirmed by the finding that 88% of sheep should receive rescue analgesia after surgery (M2) based on the Youden Index. Therefore the tool would adequately foresee that sheep were experiencing pain and help decision making to provide analgesia to improve animal welfare. Although the cut-off point may be helpful, decision making about rescue

analgesia should be taken based on clinical experience and context analysis, to ensure that sheep suffering from mild pain would be treated accordingly.

## Limitations

The current study had some limitations. The main one is that the in-person researcher edited the short videos assessed by the blind evaluators through selecting the most frequent behaviours observed in the ethogram representative of each period of observation and condensed the videos to 3 min. Although this method has been previously used to validate pain scales in cats [29], cattle [19], pigs [32] and horses [24] it is still controversial and presents advantages and disadvantages. The advantages were that because the editor was the in-person evaluator and the main author of the study (PhD student), he was the observer most familiarised with the behaviours. This guaranteed the inclusion of relevant pain behaviours. The disadvantages were that short videos might not represent the full behaviour of that particular period and in real life some behaviours may be observed only when sheep are assessed for 20 minutes. This method provides data to assess intra and inter-observer reliability and to perform all calculations for the validation of the scales and is useful to guarantee that all relevant behaviours are included in the scale development, but does not ensure that the scale is clinically applicable in real life. Another limitation is that video analysis does not necessarily equate to in-person real-time analysis. Video observation has the disadvantage of lacking some details observed in real-time, while, as an advantage it can be reviewed. A previous study in cats used the same methodology by editing short videos for initial validation of the scale [29]. The scale demonstrated validity after clinical in-person use of the instrument. Like in cats [29] the USAPS will require in-person validation to guarantee it is a valid instrument for clinical use.

The USAPS was validated only for a specific type of soft tissue surgery (abdominal—laparoscopy) and in females. Further studies are needed to test this tool in different procedures, such as orthopaedic surgery and clinical circumstances, to ensure its versatility. To establish that the instrument is valid under field conditions, clinical validation with less experienced observers is also required. Since the majority of the studies that evaluated acute pain in sheep were in lambs, this can limit the collation of data, which means the instrument needs to be tested in lambs.

Some limitations relate specifically to the videos. Although the study was blinded, some videos may have suggested the moment they were taken: at baseline, the sheep were fasting, with no available feed, hence it was difficult for the observers to interpret if the animals did not eat due to lack of food or if they really had anorexia or if they were ruminating. Around 21% of the videos at M3 were filmed at night with artificial light, which could suggest that they corresponded to M3; variations in the circadian cycle could alter some behaviours such as activity, so the reduction in activity may not be related to pain or discomfort, but to the natural reduction in activity at night [78]; given the small difference in the size of the stalls, the density of animals varied slightly, which could influence interaction and locomotion behaviours; the dark wool of some animals may also have made it difficult to evaluate some items in the videos/photos, making the analysis less accurate, especially on the facial scale.

To improve data reliability, the authors suggest that observers attend a training period, as in laboratory animals instruction and training have improved pain recognition [79].

## Conclusion

It is concluded that, after refinement of the originally proposed scale, the Unesp-Botucatu composite scale to assess acute postoperative abdominal pain in sheep (USAPS) is a valid, reliable, specific and sensitive instrument, with excellent internal consistency and discriminatory

capacity. The well-defined cut-off point for rescue analgesia supports the indication and type of analgesic therapy. To assess the clinical and experimental applicability of the scale and ensure its versatility, it is recommended that it be evaluated in other surgical procedures and in lambs.

## Supporting information

**S1 Table. Ethogram with the description of the behaviours analyzed in 48 sheep submitted to laparoscopy [5,11,20–22,34–40].**
(PDF)

**S2 Table. Criteria used to select the behaviours included in the pre-refinement of the USAPS used for video analysis (S3 Table), based on content validity and behaviours reported in the literature.**
(PDF)

**S3 Table. Pre-refinement of the USAPS to assess postoperative pain in sheep submitted to video analysis after content validation.**
(PDF)

**S4 Table. Refinement process for inclusion and exclusion of items and subitems on the USAPS.**
(PDF)

**S5 Table. Inter-observer matrix agreement of items of the USAPS, unidimensional scales and rescue analgesia indication.**
(PDF)

**S6 Table.** Scores, specificity, sensitivity and Youden index corresponding to rescue analgesia indication of the USAPS and unidimensional scales (A); 95% confidence intervals of 1,001 replications and of sensitivity and specificity >0.90 applied to estimate the diagnostic uncertainty zone of the cut-off point of each scale, according to the Youden index (B).
(PDF)

**S1 File. Data of the sheep.**
(XLSX)

## Author Contributions

**Conceptualization:** Nuno Emanuel Oliveira Figueiredo Silva, Stelio Pacca Loureiro Luna.

**Data curation:** Nuno Emanuel Oliveira Figueiredo Silva, Pedro Henrique Esteves Trindade, Alice Rodrigues Oliveira, Marilda Onghero Taffarel, Maria Alice Pires Moreira.

**Formal analysis:** Nuno Emanuel Oliveira Figueiredo Silva, Pedro Henrique Esteves Trindade.

**Funding acquisition:** Stelio Pacca Loureiro Luna.

**Investigation:** Nuno Emanuel Oliveira Figueiredo Silva, Renan Denadai, Paula Barreto Rocha, Stelio Pacca Loureiro Luna.

**Methodology:** Nuno Emanuel Oliveira Figueiredo Silva, Pedro Henrique Esteves Trindade, Renan Denadai, Stelio Pacca Loureiro Luna.

**Project administration:** Stelio Pacca Loureiro Luna.

**Resources:** Stelio Pacca Loureiro Luna.

**Software:** Pedro Henrique Esteves Trindade.

**Supervision:** Stelio Pacca Loureiro Luna.

**Validation:** Nuno Emanuel Oliveira Figueiredo Silva, Stelio Pacca Loureiro Luna.

**Visualization:** Nuno Emanuel Oliveira Figueiredo Silva, Alice Rodrigues Oliveira, Marilda Onghero Taffarel, Maria Alice Pires Moreira.

**Writing – original draft:** Nuno Emanuel Oliveira Figueiredo Silva, Stelio Pacca Loureiro Luna.

**Writing – review & editing:** Nuno Emanuel Oliveira Figueiredo Silva, Pedro Henrique Esteves Trindade, Stelio Pacca Loureiro Luna.

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
