## [Decision Letter · Decision Letter 0]

8 May 2020

PONE-D-20-07213

Validation of the Unesp-Botucatu sheep acute composite pain scale (USAPS)

PLOS ONE

Dear Prof Luna,

Thank you for submitting your manuscript to PLOS ONE. After careful consideration, we feel that it has merit but does not fully meet PLOS ONE’s publication criteria as it currently stands. Therefore, we invite you to submit a revised version of the manuscript that addresses the points raised during the review process.

We would appreciate receiving your revised manuscript by Jun 22 2020 11:59PM. To enhance the reproducibility of your results, we recommend that if applicable you deposit your laboratory protocols in protocols.io, where a protocol can be assigned its own identifier (DOI) such that it can be cited independently in the future. For instructions see: http://journals.plos.org/plosone/s/submission-guidelines#loc-laboratory-protocols

We look forward to receiving your revised manuscript.

Kind regards,

Daniel Pang

Academic Editor

PLOS ONE

Journal Requirements:

2. In your Methods section, please include a comment about the state of the animals following this research. Were they housed for use in further research?

Additional Editor Comments (if provided):

Thank you for submitting this interesting manuscript in an under-explored area of research. Both reviewers have raised important concerns regarding presentation (language, style and cohesion) and interpretation. Please take the time to provide a complete and in-depth response to their comments and suggestions.

Reviewers' comments:

Reviewer's Responses to Questions

**Comments to the Author**

1. Is the manuscript technically sound, and do the data support the conclusions?

Reviewer #1: No

Reviewer #2: Yes

2. Has the statistical analysis been performed appropriately and rigorously? 

Reviewer #1: I Don't Know

Reviewer #2: Yes

3. Have the authors made all data underlying the findings in their manuscript fully available?

Reviewer #1: No

Reviewer #2: Yes

4. Is the manuscript presented in an intelligible fashion and written in standard English?

Reviewer #1: No

Reviewer #2: Yes

5. Review Comments to the Author

Reviewer #1: Whilst I see value of this research, the current article is much too complicated and difficult to follow. There are a number of grammatical issues throughout (over use of commas) that make it difficult to read, but there are also many areas that are not fully explained - for example throughout the statistics and results section sample size changes and new estimates in sample sizes are made, but you have not made it clear why.

I have attached the paper with comments and highlighted areas that were some of the smaller details that needed to be changed. There were still many areas I haven't highlighted the you will need to consider re-writing to make them much, much clearer.

I would also suggest that you change the title - you have only validated the scale for one type of surgery (which you highlight in the limitations section) and so you can only really state that this scale can be used for this type of surgery and not all types of surgery.

Although I do understand the use of a "gold observer", there are many limitations to this and it is not really valid, especially when you then only had 3 other observers. If you are going to have one person decide, I would suggest you have a lot more people involved in the validation. In addition, rather than the observer being your gold standard, why are you simply not using the time of before and after surgery, surely this is a much better assessment of if they are or are not in pain?

Significant re-writing required to make it much clearer and much more detailed with out losing the conciseness needed.

Reviewer #2: Please see attached review document as formatting will assist in interpretation of the comments.

Here are the general comments:

The only major concern I have within the manuscript is the attempt to classify the scores produced by the scale as mild, moderate and severe pain. While I understand appreciate in post-surgical studies, other groups often compare pain scores between groups, ultimately this is a non-continuous categorical scale. The section where mild-moderate-severe is defined reads as though the authors just divided the scale into three equal sections (0-3, 4-8, 9-12), and assumed each group.

This is in stark contrast to a previous section where the score 0 is defined as “without-pain”. This within the manuscript, 0 is both no pain and mild pain. The majority of the introduction and discussion touts the robust statistical method employed in developing this scale. Something I think will be clear to the readership. This section certainly doesn’t maintain the same high standards. The study design was meant to have non-painful and painful sheep, and develop a tool which could differentiate between these two states. The intervention point correlates with the sensitivity and specificity of a “score” to classify between the two. There was no stratification, nor any clear statistical method implored to stratify these sheep with any external method. I will point out some examples in specific sections later. But notably, in figure three, the highest score with the USAPS (12, maximum pain) is related to 5/10 on the NRS, 3/4 on the SDS, <50 on the VAS, and 5/12 on the facial scale. It does not seem that maximum pain with this scale correlates with the observers’ “worst-possible pain” with the unidimensional instruments. Thus I would recommend the removal of this mild-moderate-severe classification. Further study would be required to determine any such classification.

Aside from this my other comments are minor.

I am not sure if it would be possible to shorten the introduction. While I understand the vast undertaking of the subject, I am not sure 71 references are necessary to justify the development of this scale.

6. PLOS authors have the option to publish the peer review history of their article (what does this mean?). If published, this will include your full peer review and any attached files.

Reviewer #1: No

Reviewer #2: Yes: Graeme Michael Doodnaught

---

## [Author Response · Author response to Decision Letter 0]

22 Jul 2020

ManuscriptPONE-D-20-07213: 

Validation of the Unesp-Botucatu composite scale to assess acute postoperative abdominal pain in sheep (USAPS)

Dear Editor

Thank you very much for the comments regarding this paper. All corrections have been performed according to both Reviewers and each comment responded to separately.

We hope that after these corrections you consider the manuscript suitable for publication, but we are happy to answer any further questions.

Yours sincerely,

The authors

REVIEWER: 1

Dear Reviewer:

Thank you very much for the comments regarding this paper. All corrections have been performed according to your suggestions and each comment responded to separately.

 We hope that after these corrections you consider the manuscript suitable for publication, but we are happy to answer any further questions.

 Yours sincerely,

The authors

Recommendation: 

Comments to the Author

Do you want your identity to be public for this peer review? 

Reviewer #1: No 

1. Is the manuscript technically sound, and do the data support the conclusions?

Reviewer #1: No

2. Has the statistical analysis been performed appropriately and rigorously? 

Reviewer #1: I Don't Know

3. Have the authors made all data underlying the findings in their manuscript fully available?

Reviewer #1: No

Answer: Additional data have been included (tables S5, S6, and S7) in the supporting information [S5 Table. Inter-observer matrix agreement of the USAPS, unidimensional scales, and rescue analgesia indication; S6 Table. Scores, specificity, sensitivity and Youden index corresponding to rescue analgesia indication of the USAPS and unidimensional scales (A); 95% confidence intervals of 1,001 replications and sensitivity and specificity > 0.90 applied to estimate the diagnostic uncertainty zone of the cut-off point of each scale, according to the Youden index (B); S7 Table. Residues standardized by the chi-square z-normal scale extracted from the Burt table using the USAPS scores and the SDS]. We did our best to include all data. Data on the final scale after refinement have been deposited in the system (Data of the sheep). Please let us know if anything else is required.

4. Is the manuscript presented in an intelligible fashion and written in standard English? PLOS ONE does not copyedit accepted manuscripts, so the language in submitted articles must be clear, correct, and unambiguous. Any typographical or grammatical errors should be corrected at revision, so please note any specific errors here.

Reviewer #1: No 

Answer: The manuscript has been revised by a native English speaker. Please find attached the certificate.

5. Review Comments to the Author

Please use the space provided to explain your answers to the questions above. 

REVIEWER #1: Whilst I see value of this research, the current article is much too complicated and difficult to follow. There are a number of grammatical issues throughout (over use of commas) that make it difficult to read, but there are also many areas that are not fully explained - for example throughout the statistics and results section sample size changes and new estimates in sample sizes are made, but you have not made it clear why.

Answer: The authors appreciate the effort taken to revise this manuscript. The manuscript is now more concise, and we have done our best to improve fluency. The manuscript has been revised by a native English speaker. Please find attached the certificate. More explanations have been included. The sample size was the same for all data analysis and this has been made clear.

I have attached the paper with comments and highlighted areas that were some of the smaller details that needed to be changed. There were still many areas I haven't highlighted the you will need to consider re-writing to make them much, much clearer.

Answer: Thank you for sending your comment which has been fully addressed. As mentioned above the manuscript has been rewritten.

I would also suggest that you change the title - you have only validated the scale for one type of surgery (which you highlight in the limitations section) and so you can only really state that this scale can be used for this type of surgery and not all types of surgery.

Answer: The title has been changed to “Validation of the Unesp-Botucatu composite scale to assess acute postoperative abdominal pain in sheep (USAPS).”

Although I do understand the use of a "gold observer", there are many limitations to this and it is not really valid, especially when you then only had 3 other observers. If you are going to have one person decide, I would suggest you have a lot more people involved in the validation. 

Answer: The use of gold standard data as reference has been excluded. All the statistical analysis has been performed again to include data from the two phases of all observers in all analysis.

In addition, rather than the observer being your gold standard, why are you simply not using the time of before and after surgery, surely this is a much better assessment of if they are or are not in pain? 

Answer: Please see comment above. Data from the time points before and after surgery were used to calculate responsiveness, sensitivity, and specificity.

Significant re-writing required to make it much clearer and much more detailed without losing the conciseness needed.

Answer: As mentioned before, the manuscript is now more concise; we believe it is clearer and more detailed.

REVIEWER: 1

Specific Comments

Introduction (page 2): This section could be more concise.

Answer: As mentioned before the manuscript is now more concise; we believe it is clearer and more detailed.

Page 3, Line 5 - “…only surpassed by pigs among farm animals…”

Phrasing: 

Answer: Excluded.

Page 3, Line 8 - “…ethical issues…”

Are there none in pigs and sheep? Might need rephrasing.

Answer: Excluded.

Page 4, Line 12 - “…pain dor…”

Answer: Corrected (page 3, line 26).

Page 4, Line 14 - “…to quantifies the…”

Answer: Corrected (page 4, line 1).

Page 4, Line 22-23 - “…None of these studies evaluated the criteria, content, and construct validity…”

As the author of one of these studies, I would argue that we definitely did do this!

Answer: The statistical approach was different; however we decided to exclude this to avoid misinterpretation.

Page 5, Line 18 - “…were used to stay…” “…were extrem…”

Answer: The sentence has been rephrased (page 5, lines 8-10).

Page 5, Line 23 - “…position of the cameras…”

Where did this end up being? What type of camera was involved?

Answer: Included (page 5, line 1).

Page 6, Line 10 -“…dissociative anesthesia was supplemented with 5 mg/kg of ketamine IV...”

How were you sure that these pain relief had worn off?

Answer: Please see page 5, lines 20-24.

Page 6, Line 25 - “…The local had…”

Answer: Corrected (page 6, line 9).

Page 6, Line 27 - “…using a digital camera (Gopro Hero5 Black®) positioned on a tripod...”

When? Where?

Answer: Information included (page 6, lines 11-14).

Page 7, Line 10 - “…criteria applied the scale…”

Answer: Corrected (page 7, line 5).

Page 7, Line 22 - “…To elaborate the ethogram…”

I'm not really sure what you mean here by elaborate the ethogram?

Answer: The ethogram section has been rephrased (page 7, lines 16 - 22).

Page 7, Line 22 - “…the presential researcher…”

Refer to observer (Add initials) not the presential researcher.

Answer: Included (NEOFS) (page 7, line 16).

Page 7, Line 22- “…evaluated…”

What do you mean here by "evaluate"?

Answer: Corrected (page 7, line 16).

Page 7, Line 24 – “…through recordings continuously evaluated…”

Rephrase

Answer: The ethogram section has been rephrased (page 7, lines 16 - 22).

Page 7, Line 25 – “…recognize the different behaviours…”

Construct an ethogram?

Answer: The ethogram section has been rephrased (page 7, lines 16 - 22).

Page 7, Line 25 – “…length of time…”

duration? This is an observational method and not an ethogram.

Answer: The ethogram section has been rephrased. According to Martin & Bateson (2007) (Measuring behaviour: An introductory guide. 3nd ed. Cambridge University Press; 2007) an ethogram “is ostensibly a catalogue of descriptions of the discrete, species-typical behaviour patterns that form the basic behavioural repertoire of the species”. In addition, the ethogram may be a complete list of all behaviours or it may focus on particular functional classes of behaviours (Grier, J.W. Biology of animal behaviour. Times Mirror; Mosby College Publishing, St. Louis. 1984). Although recording was not performed for 24 hours, because we had a specific interest to record pain behaviours, the time points were selected at four specific periods (pre- and postoperative, post analgesia and 24-h post). We also used this approach in previous studies (Brondani et al. Validation of the english version of the UNESP-Botucatu multidimensional composite pain scale for assessing postoperative pain in cats. BMC Vet Res. 2013;9: 1; Taffarel et al. Refinement and partial validation of the UNESP-Botucatu multidimensional composite pain scale for assessing postoperative pain in horses. BMC Vet Res. 2015;11; Oliveira et al. Validation of the UNESP-Botucatu unidimensional composite pain scale for assessing postoperative pain in cattle. BMC Vet Res. 2014;10: 1–14; Luna et al. Validation of the UNESP-Botucatu pig composite acute pain scale (UPAPS). PLoS ONE. 2020; 15(6): e0233552; Oliveira et al. Postoperative pain behaviour associated with surgical castration in donkeys (Equus asinus) Equine Vet J. 2020;10.1111/evj.13306. doi:10.1111/evj.13306).

The authors are happy to exclude “ethogram” if the Reviewer deems it necessary.

Page 7, Line 26 – “…respective percentage…”

Proportions?

Answer: Corrected (page 7, line 19).

Page 7, Line 28-29 - “…The edited videos were evaluated by four observers for the scale validation process….”

Does this not bias the results - you chose the most obvious behaviours. This would not occur in real-life?

Answer: Although the authors agree with the Reviewer´s comment, NEOFS edited the short videos to be assessed by the blind evaluators by selecting the most frequent behaviours observed in the ethogram, representative of each period of observation, and condensed the videos into 3 min. Although this method has been used previously to validate pain scales, cats [29], cattle [19], horses [24], and pigs [32] it is still controversial and presents pros and cons. 

The pros were that the editor was the most familiar observer with the behaviours because he was the in-person evaluator and the main author of the study (PhD student). This guaranteed inclusion of relevant pain behaviours. The cons were that the short videos may not represent the full behaviour of that particular period. 

This method provides data to assess intra and inter-observer reliability and to perform all calculations for the validation of the scales and is useful to guarantee that all relevant behaviours are included in the developed scale, but does not ensure that the scale is clinically applicable in real life, therefore the scale will require in-person validation as reported previously in cats [29] to guarantee it is a valid instrument for clinical use. This limitation has been included in the discussion (page 32, lines 5-18).

Page 8, Line 2 - “…manner regarding the moments…”

Make clearer.

Answer: Rewritten (page 7, lines 26 - 27).

Page 8, Line 4 - “…observers, based on their clinical experience…”

The use of "experience" does not give good validity as they are still subjective assessments.

The reviewer addresses an important issue of possible bias and, in principle, we do agree with the reviewer. However, to our knowledge, there are no better options to build the ROC curve and define the Youden index (cut-off analgesic rescue point) than by determining the scores of the scales corresponding to the indication of rescue analgesia. It is important to point out that this information was used only for ROC curve analysis and predictive criterion validity.

Page 8, Line 12 - “…in S3 Table…”

in the S3 table were used

Answer: The pain scale video evaluation has been rephrased. 

Page 8, Line 12 - “…videos, contained…”

Remove - too many commas used throughout changing the flow and meaning of the information.

Answer: The excessive commas have been removed where applicable and this section (Pain scale video evaluation) has been shortened and repositioned. We hope this paragraph is clearer now (page 7, lines 26 - 28; page 8 lines 1 - 5).

Page 8, Line 13 - “…were subdivided…”

How? Justification for the sub-divisions?

Answer: Some of the sub-items had several sub-divisions (please see S3 Table - example: item “locomotion” with a lot of grouped descriptors). They were subdivided for individual assessment of the importance of each one. This section has been simplified and we analysed the advantage of refinement in the discussion section (page 26, lines 12-16).

Page 8, Line 16 - “…totaled 33 behavioural variables…”

This is a lot! 

Answer: Please see the previous comment. This was the reason the scale was refined in order to reduce the number of variables. After refinement many subitems were excluded from locomotion (walks backwards and/or in circles), posture (kicks or stamps one or more limbs on the ground; extends one or more limbs) and miscellaneous behaviours (body tremors; crawls in ventral recumbence, without getting up).

Page 8, Line 17-18 - “…the photographic record captured by the presential observer at the end of the 20-minute recording of each moment…”

Detail of this needed. Were they stills from the video footage or taken by the observer - the effect of this on the sheep?

Answer: More information has been included before this section (page 6, lines 19 - 24). The method was the same as that described in the previous study, in which the facial pain scale was developed (McLennan et al, 2016). Although the sheep were adapted to the observer, the authors cannot guarantee that sheep were undisturbed by the presence of the observer.

McLennan KM, Rebelo CJB, Corke MJ, Holmes MA, Leach MC, Constantino-Casas F. Development of a facial expression scale using footrot and mastitis as models of pain in sheep. Appl Anim Behav Sci. 2016;176: 19–26.

Page 8, Line 19-20 - “…the gold standard observer…”

Not really valid!

Answer: As commented before, the use of gold standard data as reference was excluded. All statistical analysis has been performed again to include data from the two phases of all observers.

Page 8, Line 20 - “…greater intra-observer reliability…”

Doesn't necessarily mean they were right!

Answer: Please see previous comment.

Page 9, Line 3 - “…For all analyzes…”

Analyzis

Answer: According to our grammar consultation (https://grammarist.com/spelling/analyses-vs-analyzes) to analyse is the verb form, analysis is singular, and analyses is plural (see page 8, line 13). These terms have been corrected throughout the manuscript.

Page 9, Line 3-4 - “…considered. The sample size was estimated in 13…”

13 what? Why did you need to estimate a sample size here and not when setting up your project?

Answer: This part has been deleted as we had no previous data to estimate sample size.

Page 9, Line 7 - “…2nd assessment of the gold standard evaluator…”

When was this?

Answer: Please see previous comment.

Page 9, Line 10 “…Friedman test…”

Why?

Answer: Because the data were non-parametric (see page 8, lines 18). This information has been included in Table 1 – statistical methods (page 9, Construct validity - Responsiveness: Ethogram).

Page 10, Table 1 - Content validation “three experienced veterinarians…”

Answer: “Experienced” has been deleted (page 9, table 1).

Page 11, Table 1 Specificity “…The scores of the behavioural scale in the 2nd phase of assessment of the gold standard observer at M1 were transformed into dichotomous (level “0” - absence of pain expression behaviour for a given item; levels “1” and “2” - presence of pain expression behaviour) and applied to the respective equation…”

Relying on the observer to be corrected - why not test between before and after surgery to see if it can be correctly identified as being in each stage of the surgery, e.g. can someone correctly identify before and after surgery? This is a much better estimate of the pain. 

Answer: We believe this was the method used by assessing specificity before surgery (no pain) and sensitivity after surgery before rescue analgesia (most intense pain) and checking if each pain behaviour was absent before surgery (specificity) and present after surgery (sensitivity). The construct validity (responsiveness) presented in Table 7 and the predictive criterion validity (Table 12) confirms that the scale was able to identify pain-free animals from those feeling pain.

In addition, the use of an already validated pain assessment against your own is required 

Answer: This has been fulfilled using the criterion validity (Fig 5).

is there any evidence already to suggest the surgery is painful e.g. von-frey?

Answer: Laparotomy and laparoscopy are experimental pain models used in previous studies for several species. To our knowledge, no studies have used Von-Frey filaments to assess pain in laparotomy and laparoscopy in sheep. Laparotomy and laparoscopy increased postoperative pain scores in sheep (Zhang et al. Comparison of laparoscopic and traditional abomasal cannulation in sheep. J Vet Res 60, 113-117, 2016).

Page 11, Table 1 - Sensitivity 

Why not carry out an ROC analysis and determine your accuracy.

Answer: ROC curve analysis has already been performed. The methods for sensitivity, specificity, and ROC curve have been placed together to add clarity (see Table 1 - statistical methods - Specificity and Sensitivity (page 10).

Page 12, Table 1 “…Determination of pain intensity…” Scores were classified by intensity: low, intermediate or high, in 2nd evaluation of all evaluators at the time of greatest pain (M2).

So was this based on what observes thought?

Answer: Please see information in the third column of Table 1 (Statistical tests – Specificity and Sensitivity, page 11). A new statistical analysis has been performed to assess pain intensity, as suggested by the other reviewer.

Page 13, Line 2- “…A minimum sample size of 5 sheep [67] was estimated…”

Why is it now different? I'm not following why this keeps changing.

Answer: As previously mentioned, this has been deleted, as we had no previous data to estimate sample size.

Page 13, Line 11-12 - “…because the animals were fasting at M1 and therefore it was not possible to compare M2 vs M1 for this variable…”

Rephrasing

Answer: Corrected (page 13, lines 11-12).

Page 13, Line 14-20 - “…According to the inclusion/exclusion criteria cited for the refinement (S4 Table), the following sub-items were excluded: in the item “locomotion”, “walks backwards” and “walks in a circle”; in the item "posture", "kicks and stamps limbs on the ground" and "extends one or more limbs; in the item “miscellaneous behaviour”, “body tremors” and “crawls in ventral recumbence, without getting up” were excluded. The two sub-items that remained in the “miscellaneous” item replaced those excluded from the “posture” item. The “miscellaneous” item continued with four sub-items and was renamed “posture”. …”

This is all v. confusing. Can you not find a much simpler way of showing this?

Answer: This paragraph has been simplified (page 13, lines 15-18).

Page 13, Line 21 - “…Next, the…”

Weak transition

Answer: Corrected (page 13, line 19).

Page 14, Line 1 - “…of 48 sheep…”

Why now 48 sheep?

Answer: 48 sheep were used in the whole main study. This is now consistent throughout the manuscript - Abstract (page 1, line 25) and Material and Methods (page 5, line 2; page 6, line 14).

Page 14, Line 2 - “…total time of each evaluation moment (20 mins)…”

But you cut them down to 3 minutes?

Answer: This table contains the proportion of behaviours expressed in the 20 min videos as described in methods used to perform the ethogram (page 7, lines 13-19). After the ethogram was completed the videos were edited for inclusion of the predominant behaviours for a period of about three minutes at each moment (page 8, lines 19-21).

Page 19, Line 3 - “…192 sheep for all evaluators…”

Why now 192 sheep?

Answer: The number of sheep was originally based on 48 sheep for each observer and 192 sheep for all evaluators. This has been deleted to avoid misinterpretation. 

Page 20, Line 16 - “…Fig 5. Biplot for…”

Box plot?

Answer: This is a biplot. Biplots are a type of exploratory graph used in statistics, a generalization of the simple two-variable scatterplot. A biplot is a representation of both samples and variables of a data matrix displayed graphically. Samples are displayed as points while variables are displayed either as vectors, linear axes, or nonlinear trajectories.

Page 22, Line 11 - “…of the ROC curve…”

Full abbreviation given here, not in figure legend.

Answer: Corrected (table 1 - Statistical test, page 10).

Page 28, Line 4-5 - “…As this is not the case in sheep, since there is no validated scale with robust statistics to assess postoperative pain…”

I think you need to make it clear - for this context as there are validated gold standard measurements of pain in other context for sheep e.g. lameness and footrot and the use of von-frey filaments to assess hyperalgesia. 

Answer: This part has been deleted to avoid misinterpretation as commented before at the introduction section. According to our knowledge, there are no studies with the use of Von-Frey filaments to assess acute pain (in laparotomy and laparoscopy) in sheep.

Page 29, Line 2-4 - “…Otherwise when comparing the proposed scale with the facial scale in sheep [69], which, although not fully validated, was considered for comparison, the correlation was moderate…”

You could mention here that the facial scale for 69 may not be suitable for the sheep in Brazil - although suffolk sheep (floppy) ears were sued, the scale was limited with regards to different breeds, and so may not be suitable, hence the low correlation.

Answer: This has been included as suggested (last sentence of this paragraph) (page 28, lines 25-26).

Page 33, Line 18 - “…acute postoperative pain…”

Given only one type of operation performed, would it not be better to state the type of surgery instead - you cannot state all acute operations as it hasn't been tested in these. 

Answer: Corrected to: “…assess acute postoperative abdominal pain in sheep (USAPS)…” (page 33, line 23).

Manuscript PONE-D-20-07213: 

Validation of the Unesp-Botucatu composite scale to assess acute postoperative abdominal pain in sheep (USAPS)

Dear Editor

 Thank you very much for the comments regarding this paper. All corrections have been performed according to both Reviewers and each comment responded to separately.

 We hope that after these corrections you consider the manuscript suitable for publication, but we are happy to answer any further questions.

 Yours sincerely,

The authors

REVIEWER: 2

Dear Reviewer:

Thank you very much for the comments regarding this paper. All corrections have been performed according to your suggestions and each comment responded to separately.

 We hope that after these corrections you consider the manuscript suitable for publication, but we are happy to answer any further questions.

 Yours sincerely,

The authors

Recommendation: 

Comments to the Author

Do you want your identity to be public for this peer review? For information about this choice, including consent withdrawal, please see our Privacy Policy.

Reviewer #2: Yes: 

Comments to the Author

1. Is the manuscript technically sound, and do the data support the conclusions?

Reviewer #2: Yes

2. Has the statistical analysis been performed appropriately and rigorously?

Reviewer #2: Yes

3. Have the authors made all data underlying the findings in their manuscript fully available?

Reviewer #2: Yes

4. Is the manuscript presented in an intelligible fashion and written in standard English?

Reviewer #2: Yes

-----

5. Review Comments to the Author

Please use the space provided to explain your answers to the questions above. 

Reviewer #2: Please see the attached review document as formatting will assist in interpretation of the comments.

Here are the general comments:

I would like to commend the authors who present a robust statistical study to validate an acute pain scale in sheep.

Answer: The authors appreciate the time taken to review this manuscript and for including your suggestions that have been fully considered unless otherwise stated. 

The statistical analysis has been performed again according to Reviewer 1 (data from all observers and phases 1 and 2 were combined instead of using data from phase 2 of the gold standard observer). Although small changes were observed in the Tables presented in the results section, the results are very similar to the previous submission. All changes are highlighted in yellow.

The only major concern I have within the manuscript is the attempt to classify the scores produced by the scale as mild, moderate and severe pain. While I understand appreciate in post-surgical studies, other groups often compare pain scores between groups, ultimately this is a non-continuous categorical scale. The section where mild-moderate-severe is defined reads as though the authors just divided the scale into three equal sections (0-3, 4-8, 9-12), and assumed eachgroup.

This is in stark contrast to a previous section where the score 0 is defined as “without-pain”. This within the manuscript, 0 is both no pain and mild pain. The majority of the introduction and discussion touts the robust statistical method employed in developing this scale. Something I think will be clear to the readership. 

This section certainly doesn’t maintain the same high standards. The study design was meant to have non-painful and painful sheep, and develop a tool which could differentiate between these two states. 

The only major concern I have within the manuscript is the attempt to classify the scores produced by the scale as mild, moderate and severe pain. While I understand appreciate in post-surgical studies, other groups often compare pain scores between groups, ultimately this is a non-continuous categorical scale. The section where mild-moderate-severe is defined reads as though the authors just divided the scale into three equal sections (0-3, 4-8, 9-12), and assumed each group.

This is in stark contrast to a previous section where the score 0 is defined as “without-pain”. This within the manuscript, 0 is both no pain and mild pain. The majority of the introduction and discussion touts the robust statistical method employed in developing this scale. Something I think will be clear to the readership.

This section certainly doesn’t maintain the same high standards. The study design was meant to have non-painful and painful sheep, and develop a tool which could differentiate between these two states.

Answer: The authors thank you for pointing this out. Two consultant statisticians were hired, and a new method used to present this analysis appropriately. We hope that this analysis now fulfils your expectation, otherwise please let us know.

In this study it was expected that sheep would not suffer pain at the preoperative point (M1), followed by a maximum expression of pain at the immediate postoperative moment (M2), reduction in pain after analgesia (M3), and little pain 24h after surgery (M4). We consider that the instrument for assessing pain does not only contemplate the diagnosis of pain but also quantifies pain intensity. Therefore, we used the simple descriptive scale (SDS) as a reference tool for pain intensity because SDS describes four levels of pain (1 = no pain; 2 = mild pain; 3 = moderate pain; and 4 = severe pain). A multivariate statistical analysis was applied to explore correspondences between categorical variables to classify the total score of USAPS in each of the SDS pain intensities. After consulting two statisticians, correspondences were calculated by two methods: (1) visual judgment of the proximity of the USAPS and SDS levels in the perceptual map by multiple correspondence analysis (MCA); and (2) value of the standardized chi-square residue. First, both scales were submitted to MCA which uses the basic concept of chi-square to standardize the frequencies of the variables and to form a basis for correspondences (interrelationship) between categorical variables (Greenacre, 2010). In addition, to make the USAPS classification more accurate, the correspondences between the scale levels by the chi-square residue were used to group the USAPS levels among each level of pain intensity according to the SDS. 

To confirm a significant difference between the USAPS intensities according to MCA, the Scott Knott test was applied. This test uses a “likelihood” relation to judge the difference between clusters. In view of this new approach, we understand that the classification of USAPS intensities was carried out by robust and appropriate mathematical methods, already applied in other areas to group and classify categorical variables.

Salas, Y., Márquez, A., Diaz, D., & Romero, L. (2015). Epidemiological study of mammary tumors in female dogs diagnosed during the period 2002-2012: a growing animal health problem. PloS one, 10(5).

Wilfart, A., Espagnol, S., Dauguet, S., Tailleur, A., Gac, A., & Garcia-Launay, F. (2016). ECOALIM: A dataset of environmental impacts of feed ingredients used in French animal production. PloS one, 11(12).

Guinet, F., Avé, P., Jones, L., Huerre, M., &Carniel, E. (2008). Defective innate cell response and lymph node infiltration specify Yersinia pestis infection. PloS one, 3(2).

Le Rumeur, E., Carre, F., Bernard, A. M., Bansard, J. Y., Rochcongar, P., & De Certaines, J. D. (1994). Multiparametric classification of muscle T 1, and T 2 relaxation times determined by magnetic resonance imaging. The effects of dynamic exercise in trained and untrained subjects. The British journal of radiology, 67(794), 150-156.

The intervention point correlates with the sensitivity and specificity of a “score” to classify between the two. There was no stratification, nor any clear statistical method implored to stratify these sheep with any external method. I will point out some examples in specific sections later.

Answer: The stratification of the sheep with and without pain was not performed before the statistical analysis to estimate pain intensity for two reasons. Stratification based on the assessment time points could be ambiguous, because even in the immediate postoperative period (M2) some sheep did not appear to be suffering pain, possibly due to the individuality of the pain sensation. Besides this, the stratification by the cut-off point was not performed because by using this new mathematical approach sheep without pain (1 = no pain) and with pain (2 = mild pain; 3 = moderate pain; and 4 = severe pain) were included in the SDS pain intensities and classified below or above the cut-off point of both scales (≤ 3for USAPS and ≤1 for SDS), which shows consistency with the mathematical method used for the classification of pain and with the observers' scores; as well as which the intensity "no pain" is relevant and must be computed. This was the reason the statistical analysis was performed with all-time points and all observers to define the intensity of the pain.

But notably, in figure three, the highest score with the USAPS (12, maximum pain) is related to 5/10 on the NRS, 3/4 on the SDS, <50 on the VAS, and 5/12 on the facial scale. It does not seem that maximum pain with this scale correlates with the observers’ “worst-possible pain” with the unidimensional instruments. Thus I would recommend the removal of this mild-moderate-severe classification. Further study would be required to determine any such classification.

Answer: Unidimensional scales (NRS, SDS, and VAS) depend on the expertise of the evaluator and are considered more subjective when compared to composite and multidimensional scales. Because composite scales have well-defined descriptors, they suffer less influence of observer's interpretation and are usually more reliable. However, all these scales follow the same logic in which the highest numerical score represents the highest level of pain. Therefore, it is expected that the scores of unidimensional scales are not identical to USAPS, but are sufficiently correlated with the USAPS (rho> 0.81). 

We recognize that this approach has limitations. Our group published a scale in pigs using the same approach (Luna et al 2020. Validation of the UNESP-Botucatu pig composite acute pain scale (UPAPS). PLoS ONE, 15(6), e0233552) and we hope to improve the quantification of pain in the future, but it is the best we can do currently according to the statisticians we consulted. If after this new analysis the reviewer is not satisfied, the authors are happy to remove this analysis of pain intensities.

Aside from this, my other comments are minor.

I am not sure if it would be possible to shorten the introduction. While I understand the vast undertaking of the subject, I am not sure 71 references are necessary to justify the development of this scale.

Answer: We have done our best to shorten the manuscript without leaving out necessary information. The number of references in the introduction has been reduced by almost half and the manuscript has been reviewed by a native English speaker to improve fluency.

Specific Comments

Page 1, Line 24 – “…is essential to diagnose pain and guarantee effective…”

No scale can guarantee analgesia. This term should be removed.

Answer: Corrected to “is essential to diagnose pain and improve decision making for analgesia” (page 1, lines 22-23).

Page 3, Line 7 – “…and teaching, as their limitations to using these other species as models…”

Consider: “…and teaching, as there are limitations to using these other species as models…”

Answer: This has been deleted to reduce the size of the introduction.

Page 3, Line 9 – “…studies on osteoporosis [17] and bone regeneration and osteointegration of dental implants [18].”

Consider: “…studies on osteoporosis [17], bone regeneration and osteointegration of dental implants [18].”

Answer: This has been deleted to reduce the size of the introduction.

Page 3, Lines 11-15 – “Although there are several useful indicators to assess nociception in experimental situations, such as the injection of formalin into the interdigital space [19], von Frey filaments [20], tourniquets [21], electrical stimuli [22], and pneumo-mechanical stimulus in limbs [23], these are not reproducible and are difficult to use in clinical situations.”

Are the authors trying to reference the instruments used to measure pain in references 19-23?

Because the following are the

[reference] – pain model / measurement technique of each:

[19] – Interdigital formalin injection / Limb withdrawl and behaviouralassessment

[20] – Peroneal nerve injury / von Freyfilament

[21] – Tourniquet / Fractal HR

[22] – Electrical stimuli /EEG

[23] – Pneumo-mechanical stimulus of the limbs / Limb withdrawl and breath-to-breathCO2 

However the bold are what is reported in the manuscript. Please clarify what is meant as “indicator to assess nociception” and adjust accordingly. 

I swapped von frey filament for the pain model to make everything coherent: such as interdigital formalin injection [19], peroneal nerve injury[20], tourniquets [21], electrical stimuli [22], and pneumo-mechanical stimulus of the limbs [23],

Answer: The paragraph has been shortened to reduce the size of the introduction and the references have been updated “Although there are several experimental methods to assess nociception [10-14]” – see page 3, line 1.

Page 3, Line 15: “Actigraphy can be used to monitor…”

The authors should define actigraphy as it is not something the readership might be familiary with, nor is the term used in the reference provided.

Answer: Corrected to “Actigraphy can be used to monitor the sheep activity” (page 3, lines 2-3).

Page 4, Line 12: “However, these instruments exclusively evaluate the intensity of pain dor…” 

Consider removing the word “dor”

Answer: Corrected (page 3, line 26).

I am not sure this sentence also fully describes the advantage of multidimensional measurement over unidimensional instruments. Giving an example of how a tooth ache vs visceral pain cannot be simply compared might be another approach.

Answer: Corrected to (in bold) “However, these instruments exclusively evaluate the intensity of pain, whereas multidimensional or composite scales include sensory, motor, and emotional qualities and may be developed to differentiate specific types of pain” – page 3, lines 27-28.

Page 4, Line 14: “…ethogram is produced to quantifies the duration…”

Consider: “…ethogram is produced to quantify the duration..”

Answer: Corrected (page 4, lines 1).

Page 4, Line 23: “To guarantee the reliable measurement of pain…”

Again no scale can guarantee measurement. Consider: “To improve the reliability of pain measurement” or something similar.

Answer: Corrected as suggested (page, 4, line 9-10).

Page 5, Lines 1-5: “Given the hypothesis that the scale proposed in the current study presents reliability, and content, construct and criterion validities, the main objective of this study was to validate a behavioral scale to assess acute pain in sheep undergoing soft tissue surgery(laparoscopy), constructed from the literature and an ethogram, followed by refinement and subsequent validation, with definition of the cut-off point for analgesic intervention.”

This is quite a long run-on sentence. Consider dividing it up for improved clarity. Consider: “The main objective of this study was to validate a behavioural scale to assess acute pain in sheep undergoing soft tissue surgery (laparoscopy). The authors constructed an ethogram from the literature, then used videos from this study for further refinement, and to define a cut-off point for analgesic intervention. The authors hypothesize that the final scale produced in the current study will be reliable and demonstrate content, construct and criterion validities.”

Answer: Corrected as suggested (page 4, lines 19-24).

Page 5, Lines 17-21: “The sheep were placed in stalls, close to the pen they lived in and where they were used to stay like a shelter when atmospheric conditions were extrem, 24 hours before the start of the study, during which they fasted for feed, and for 12 hours they fasted for water. In each stall (3x2x1.10mor2.20x2x1.20m-lengthxwidthxheight) 6 to 8 sheep or 2 to 4 sheep were housed, respectively.”

These two sentences have a few typographical errors and lack clarity. 

Consider: “During the study period, sheep were housed in large (3 x 2 x 1.1m, length x width x height) and small (2.2 x 2 x 1.2m) pens with 6 to 8 or 2 to 4 animals each, respectively. The sheep were habituated to the pens for 24 hours before the start of the study.”

Answer: Corrected as suggested (page 5, lines 6-10).

Page 5, Line 23: “…position of the cameras and other adjustments, in order to guarantee the quality of filming.

Consider: “…positioning of the cameras and other adjustments, in order to optimize the quality of filming.”

Answer: Corrected as suggested (page 5, line 1-2). The other reviewer also asked for inclusion of the camera model.

Page 6, Lines 6-7: “…and anesthetic infiltration with 2% lidocaine without vasoconstrictor (Xylestesin…”

The dose used for incisional block is missing.

Answer: The information has been included (page 5, lines 19-20).

Page 6, Lines 12-13: “In all animals, the same experienced surgeon performed a laparoscopy for follicular aspiration and replacement of follicular cells [73–75]…”

Reference 73 is for ovariectomy by laparotomy, partial video assisted ovariectomy and total laparascopic ovariectomy. Not sure the relevance here.

Answer: Only one reference has been left here as it was used later as well (page 5, line 26).

Page 6, Lines 18-19: “Fig 1.”

Is this figure actually necessary? The order of events isn’t very complicated. 

Answer: This has been maintained to add clarity for the other reviewer (page 6, line 16-17).

Also this figure doesn’t mention incisional lidocaine.

Answer: Included (page 6, Fig 1)

Page 6, Lines 22-23: “The procedures started at 9 am and the evaluations of the last animals ended around 7 pm.”

Consider adding after this sentence that the 24 hour measurement occurred the next day.

Answer: Included (page 6, lines 6-7).

Page 6, Line 24: “…and the mean temperature and humidity varied between…”

Was this the mean high? Or mean daily temperature? Please add this descriptor.

Answer: Corrected to mean daily temperature,

Page 6, Line 26: “The presential observer made the recordings…”

“presential” is used multiple times in the manuscript, and should be removed. 

Consider: The same observer (author’s initials) made the recordings…”

Answer: Corrected to “the in-person observer (NEOFS) made the recordings…” (page 6, line 19).

Page 7, Lines 1-2: "The camera was turned on and the presential researcher left the place and stayed at least 10 m in order to minimize human interference in the behavior of the sheep."

Consider: “The observer turned on the camera and then distanced themselves at least 10 m from the pens in order to minimize the effect of human presence on the behaviour Answer: Corrected according to the Reviewer´s suggestion (page 6, line 20-22). 

Page 7, Line 22: “To elaborate the ethogram, the presential researcher evaluated…” Consider: “For further elaboration of the ethogram, the same observer who recorded the videos evaluated….”

Answer: This paragraph has been reformulated to better explain the elaboration of the ethogram to the other reviewer (page 7, lines 15-21).

Page 8, Line 18: “…the presential observer…” 

Remove the word presential.

Answer: Corrected (page 7, line 15).

Table 1:

Per my previous comment, I would recommend removing the determination of pain intensity. However if the authors justify its inclusion, the classification in this table is low, intermediate and high, not mild-moderate-severe

Answer: According to the new analysis the classification has been maintained as described by the simple descriptive scale, because SDS was the reference to quantify the intensity of pain by the USAPS. However, the authors are happy to change this if the reviewer is not satisfied (page 9, Table 1). 

Page 13, Line 2: “A minimum sample size of 5 sheep [67] was estimated.”

This seems out of place. The 48 sheep reported in the methods should be stated here, and the sample size estimate should be in the methods.

Answer: The sample size calculation has been removed as we had no previous data to estimate sample size.

Page 13, Line 23: “ranging from zero (without pain) to 12 (maximum pain).” 

This is related to my previous comment. This statement treats the USAPS as a unidimensional scale. I appreciate later that the values were treated as unidimensional for mathematical/statistical reasons, but this doesn’t seem appropriate.

Answer: A paragraph in the discussion raises the debate about the mathematical and biological dimensions (page 27, lines 14-23). The authors agree that the scale was multidimensional in biological terms.

Figure 6

There should be a definition of what blue/orange/grey means in the figure description

Answer: Included in the legend (new figure 3 - page 16, line 8).

Page 23, Line 12: “10 The percentage of animals present in the diagnostic uncertainty zone (scores 3 and 4) was low at all times for all evaluators (11%; 9 - 15%). At M2, this percentage for all evaluators grouped was 7% (0 to 13%), which ensures that 93% of the sheep were detected as suffering pain with confidence at this moment (Table 13).”

Looking at the box plot in figure 4, there are values less than 3. So 93% (100 minus 7%) of sheep were not detected as painful. Eyeballing there looks like there are 5 animals with a score of less than 3.

So the instrument detected 7% in a grey area and% as painful and _% as non-painful. The 93% is the combined “clinically clear or useful”.

Answer: Please accept our apologies for this mistake and lack of attention. We are grateful to the reviewer for pointing this out. This sentence has been deleted: “At M2 95% (92-99) of the sheep were detected as suffering pain with confidence at this moment” (Table 11).

Page 24, Lines 1-20

 Per my previous comment, I do not see the value of this section.

The authors have seemingly defined pain intensity as mild (0-3), moderate (4-8) and severe pain (9-12). Then performed cluster analysis at M2

Answer: Please check previous comments about this analysis. Data from all time-points are now used. If the reviewer is not satisfied with the new more robust statistical analysis, the authors are happy to remove this section from the methods, results, and discussion.

Based on the assumption from the study design, this is the painful phase, and “clinical experience” of the video evaluators suggests about 90% of these animals need rescue. But there was no stratification applied by any external measure. So, the “intensities” here have simply been defined by the authors’ arbitrary cut-offs. Because, as mentioned, the “12’s” produced with this scale are not associated with the highest scores of the unidimensional scales.

Answer: Please see the previous comment about the same subject.

The authors expected M4 to be worse than or equal to M3. Similar to the previous ruminant scale. Perhaps the model of pain used in this study doesn’t achieve extremes of intensities. The ovariectomy reference #73 from the surgical description, using an NRS shows scores of 0.3 for video assisted or laparoscopic surgery (vs 5 for laparotomy). This scale may be sensitive to lower pain-intensities associated with minimally invasive surgeries. Future study, as the authors suggest later, is warranted.

Answer: Thank you for this comment. The authors are currently performing a study with orthopaedic surgery to clinically validate (or not) this scale.

Additionally: Figure 9’s description mentions a>b>c, but there are no letters in the figure. Answer: Corrected, the letters have been included (page 23, Fig. 9).

Page 24, Line 27: “…as it can guarantee that sheep benefit from analgesia when necessary…” Again, the scale cannot guarantee. This word should be changed. Even the authors recommend later in the discussion that clinical evaluation should still be considered for scores <4.

Answer: Corrected (page 24, lines 3).

Page 26, Line 17: “…the trans-operative period.”

Do the authors mean peri-operative period? I am not sure what the trans-operative period is.

Answer: This has been deleted because it was meant to say that restraint in the preoperative moment may lead to vocalization (page 25, line 20).

Page 26, Line 19: “Thus, vocalization is not, in the sheep species, an indicator of postoperative pain.”

Considering the great length, the authors go into in the introduction to talk about the limited availability of pain assessment in sheep in the literature. And that the references, as later discussed are mostly in lambs. The authors surely cannot claim this study as definitive proof that vocalization is not a component of pain, considering the current study only used one pain model.

This statement should be revised.

Answer: Corrected to “Thus vocalization is not an indicator of postoperative pain in adult sheep submitted to laparoscopy, as reported previously in sheep undergoing severe painful stimuli” (page 25, lines 21-22).

Page 26, Line 21: “A differential of the current study…” 

I am not sure what is meant by differential? Do the authors mean “A major difference in the current study compared to the literature…”?

Answer: Corrected (page 26, line 24).

Page 27, Line 23: “…it is premature to conclude about the dimension of the proposed pain…” I am not sure “dimension” is what is meant here.

Is the comment meant to be about the universal application to other pain types?

Answer: This was focused only on the mathematical approach and corrected to differentiate the mathematical and biological approaches (see page 3, lines 25-28). The last sentence of this paragraph has been changed to exclude the relationship between dimensions and types of pain (page 27, lines 22-23).

Page 28, Line 4: “As this is not the case in sheep…”

This statement, following the previous sentence, reads as though it is not necessary to compare the new instrument with a gold standard. Rather than the intended: because no gold standard exists, there is no standard to compare the current instrument with.

Consider removing this statement, and writing: “…considered the gold standard [71]. Since there is no validated scale with robust statistics to assess postoperative pain in sheep…”

Answer: The statistical analysis has been performed again as suggested by the other reviewer. As data from all observers were grouped there are no longer comparisons against the gold standard observer. The paragraph has been rephrased as suggested by the other reviewer.

Page 29, Line 9: “…Youden Index after surgery (M1) rescue analgesia was indicated in 93% of sheep…”

This should be M2, which is the moment after surgery.

Answer: Corrected (page 31, lines 21-22).

Per my previous comment, this wasn’t 93% of sheep that needed rescue analgesia. The 7% are the sheep with scores 3-4, and 93% with scores <3 or >4. And there are (from figure 4) at least 5 with scores less than 3 (non-painful).

Answer: Please see the previous comment about the same subject.

Page 29 Line 10-11: “…therefore the tool would foresee well that sheep were undergoing pain and then be treated, guaranteeing the animals’ welfare.

The instrument isn’t perfect, thus it cannot guarantee the animals treatment of pain, and also pain is only one component of welfare. 

Consider: “…therefore the instrument is a

Answer: Corrected (page 31, lines 21-24).

Page 29, Lines 17-21: “In this study, the differences observed in the pain scores between the moments, and especially at the expected moment of greatest pain compared to the other moments, confirm that the proposed scale is responsive both to identify intense degrees of pain, as well as moderate degrees, which occurred after rescue analgesia, or even mild pain, which occurred 24 hours after surgery.”

Per my previous comment

• The authors expected M2>M4≥M3>M1 for pain scores, but actually observed M2>M3>M4>M1

• Thus the pain model does not provide the same order of scores as in previous studies in other species, or this instrument respondsdifferently.

Answer: This part has been amended (page 29, lines 6-9).

• Regardless, this order does not confer statistically robust stratification and cannot then be used to assume mild-moderate-severe pain classifications for a categoricalscale.

• The design was not to achieve no pain → surgical pain → mild pain → moderate pain. This is impossible to predict. The authors developed a scale which differentiates between M1 and M2 (non-painful and painful) and then proves responsive to the treatment with rescue analgesia in M3. And then the pain had probably subsided due to the model by the 24hevaluation.

• Also, with the given criteria, a score of 0/12 on the scale would be “mild pain”. Though the authors frequently switch between mild and no-pain for 0/12.

Answer: Please see the previous comment about the same subject. The manuscript has been reformulated and so has this analysis. If after this new analysis the reviewer is not satisfied, the authors are happy to remove this analysis of pain intensities

Page 29, Line 28: “…apparently the instrument can be used in different sheep breeds.” Considering only 3 breeds, all of which are dairy sheep from the same geographical region, were assessed. It is rather presumptuous to state the scale can be used on all sheep breeds. The authors can postulate this theory, but should recommend further study before making such a definitive statement.

Answer: We agree with the reviewer. However, with the new statistical analysis, there was a difference between species, so we have excluded this sentence.

Page 32, Line 3-7: “To our knowledge, the scales that assess acute pain in various animal species do not classify pain intensity based on their scores, except in an empirical way [44]. In this study, the zone of diagnostic uncertainty (3 - 4) corresponded to the lower limit of moderate pain scores (4), insuring that sheep suffering from moderate pain would be treated according to the cut-off point.”

Per my previous comment

I would imagine other scales were not classified in this manner because they are also non- continuous categorical scales used to assess whether or not pain is present.

This paragraph also highlights the completely arbitrary definition of “moderate pain” WSAVA Guidelines for the recognition, assessment, and treatment of pain describes the perceived pain of ovariohysterectomy in dogs and cats as “Moderate”.

I fully appreciate this is perceptions, and species differences can be vast. However:

The same references used in the surgical description (73) states mean pain score of ovariectomy by laparotomy as 5.6 (on a 10-pt scale, 0-9) and 0.3 out of 9 for both video- assisted and pure-laparoscopic ovariectomy.

Additionally, suggesting that this pain scale recommends that the readership should not treat “mild pain” (scores 0-3) takes away from the robust statistical approach of all the other steps in this instruments development.

I do not understand what this classification achieves.

Answer: Please see the previous comment about the same subject. Please also see the paragraph of discussion before Limitations (page 31, lines 25-27; page 32, lines 1-2).

Page 32, Line 17: “…video analysis does not necessarily equate to presential analysis.” 

Consider: “…video analysis does not necessarily equate to in-person real-time analysis.”

Answer: Corrected to“…video analysis does not necessarily equate to in-person real-time analysis” (page 32, lines 18-19).

Page 32, Line 22: “According to a study by…”

Consider: “According to studies by our group…” since multiple references are used.

Answer: This part has been rephrased.

---

## [Decision Letter · Decision Letter 1]

25 Aug 2020

Validation of the Unesp-Botucatu composite scale to assess acute postoperative abdominal pain in sheep (USAPS)

PONE-D-20-07213R2

Dear Dr. Luna,

We are pleased to inform you that your manuscript has been judged scientifically suitable for publication and will be formally accepted for publication once it meets all outstanding technical requirements.

Within one week, you will receive an e-mail detailing the required amendments. When these have been addressed, you’ll receive a formal acceptance letter and your manuscript will be scheduled for publication.

Kind regards,

Daniel Pang

Academic Editor

PLOS ONE
---

## [Author Response · Author response to Decision Letter 1]

2 Sep 2020

RESPONSE TO EDITOR 

Dear Editor

Thank you very much for your comments and suggestions regarding this paper. All corrections have been performed according to Editor and Reviewer´s comments and each of them was answered separately. 

We hope that after these corrections you consider the manuscript suitable for publication, but we are happy to answer any further questions.

Yours sincerely,

The authors

Additional Editor Comments (if provided):

Thank you for addressing the majority of the reviewer comments. A few fairly minor issues remain.

General Comments:

I thank the authors for their amendments to the manuscript. Many sections read better now and much of what was performed is clearer.

My main concern remains with the intensities or degrees of pain. I appreciate the subjectivity of the SDS, and even in the study the authors found interobserver differences in the scores at each of the moments, even differences based on breed. As mentioned by the authors there is a high correlation between the USAPS and all subjective scales (facial scale being the exception there). So it is not a surprise that cluster analysis would yield similar results, particularly when observers are performing pain assessments with these scales simultaneously, introducing bias.

But again, the authors have not justified why the divisions for intensity are no pain (0-2), mild pain (3-6), moderate pain (7-9) and severe pain (10-12). These seem to merely to provide equal divisions to maintain proportionality to the SDS.

In the discussion the authors even make arguments which undermine these divisions:

- The discussion repeatedly mentions the limitation of assessing appetite. But since the youden index and AOC remained the same with and without appetite, the intervention point for differentiating “painful” and “non-painful” animals remains the same. The authors state the scale is valid whether appetite is used or not. But by not including appetite, this will change the scores from a maximum of 12 to a maximum of 10. Effectively eliminating all but one value which would be considered “Severe pain”

- The authors also state the importance of the predictive criterion, and the low percentage of animals in the diagnostic uncertainty zone as being signs this scale is valid and clinically relevant. However the last paragraph of the discussion states that animals experiencing “mild pain” (3-6) should possibly receive analgesia, despite the cut-off being 4. I appreciate that clinically no scale is perfect, and even the USAPS would erroneously suggest that almost 1/3rd of sheep pre-surgically requires analgesia. These definitions of “mild pain” undermine the arguments for the robust validity of the USAPS scale.

For these reasons I still do not see the value in this section for an already long and involved manuscript. The study design was intended for the development of the pain scale and producing a reliable and valid pain scale. In its current state, basing pain intensities on the SDS as the sole manner of associating pain intensity seems inappropriate, especially when the scale would define almost 1/3 of animals as experiencing “mild pain” at M1.

Answer: The authors appreciate your comments and the section about pain intensities was removed from the manuscript.

Apart from this I only have a few specific points which are mentioned below.

Specific Comments

Answer: Thank you very much for the comments regarding this paper. All corrections have been performed and each comment responded to separately.

Page 3 Line 1-2:

“…several experimental methods to assess nociception [10-14] they are not reproducible and are…”

I would consider rephrasing this.

If the nociceptive technique is not reproducible, then the technique is not valid even experimentally. This is very different from something that is unreliable in unhabituated clinical patients.

Answer: The sentence has been rephrased as suggested (page 2, line 27).

Page 6 Line 22: “NEOFS approached…”

This should be simply the observer, as that person has already been defined as NEOFS

Answer: Corrected (page 6, line 19).

Page 7 Line 15: “The observer NEOFS…”

NEOFS should be in brackets

Answer: Corrected (page 7, line 12).

Page 7 Line 16: “NEOFS watched…”

This should be “The observer” as they have already been defined as NEOFS

Answer: Corrected (page 7, line 13).

Table 3 – Head Position: “Occipital above withers…”

“Head” position is used repeatedly in the manuscript, and “occipital” is only used in this Table.

For clarity replace Occipital with Head

Answer: Corrected in this Table and also in the S1, S2 and S3 tables of the supporting information. Occipital was maintained only in the behaviour descriptor of S1 Table only to make clear that assessments were based on the occipital region.

RESPONSE TO REVIEWER 1

Dear Reviewer:

Thank you very much for the comments regarding this paper. All corrections have been performed according to your suggestions and each comment responded to separately.

 We hope that after these corrections you consider the manuscript suitable for publication, but we are happy to answer any further questions.

 Yours sincerely,

The authors

Reviewers' comments:

Reviewer's Responses to Questions

Comments to the Author

1. If the authors have adequately addressed your comments raised in a previous round of review and you feel that this manuscript is now acceptable for publication, you may indicate that here to bypass the “Comments to the Author” section, enter your conflict of interest statement in the “Confidential to Editor” section, and submit your "Accept" recommendation.

Reviewer #1: All comments have been addressed

2. Is the manuscript technically sound, and do the data support the conclusions?

Reviewer #1: Yes

3. Has the statistical analysis been performed appropriately and rigorously?

Reviewer #1: Yes

4. Have the authors made all data underlying the findings in their manuscript fully available?

Reviewer #1: Yes

5. Is the manuscript presented in an intelligible fashion and written in standard English?

Reviewer #1: Yes

6. Review Comments to the Author

Reviewer #1: I applaud the authors for making the manuscript much clearer and much more concise, except for the discussion which still needs some work on improving the narrative and flow of the information. 

Answer: The authors appreciate the effort taken to revise this manuscript. The manuscript is now more concise, and we have done our best to improve fluency. The pain intensity section was removed from all sections according to the Editor´s comments.

There are too many very short paragraphs and 'appetite' appears in many of them doted throughout the writing. This is the only section that really needs work. 

Answer: The discussion is now more concise and clear. The comments about appetite were condensed.

I would also change figure 6 so that it doesn't have the colour 'blobs' around the boxes - I don't think they are helpful in getting your points across.

Answer: Corrected

In the limitations section, rather than 'pros' and 'cons' I would stick to advantages and disadvantages as proper, academic English. 

Answer: Corrected (page 30, lines 3 and 6).

In addition on line 22 of the limitations you state "The scale was proven..." - I never like "proven" in scientific language as science never "proves" anything - consider instead 'demonstrate' or 'showed' etc.

Answer: The sentence has been rephrased (page 30, lines 16).

Otherwise - well done, much easier to read than before and see exactly what a good piece of work has been carried out.

---

## [Editor Report · Decision Letter 2]

10 Sep 2020

Validation of the Unesp-Botucatu composite scale to assess acute postoperative abdominal pain in sheep (USAPS)

PONE-D-20-07213R2

Dear Dr. Luna,

We are pleased to inform you that your manuscript has been judged scientifically suitable for publication and will be formally accepted for publication once it meets all outstanding technical requirements.

Within one week, you will receive an e-mail detailing the required amendments. When these have been addressed, you’ll receive a formal acceptance letter and your manuscript will be scheduled for publication.

Kind regards,

Daniel Pang

Academic Editor

PLOS ONE
---

## [Editor Report · Acceptance letter]

30 Sep 2020

PONE-D-20-07213R2 

Validation of the Unesp-Botucatu composite scale to assess acute postoperative abdominal pain in sheep (USAPS) 

Dear Dr. Luna:

I'm pleased to inform you that your manuscript has been deemed suitable for publication in PLOS ONE. Congratulations! Your manuscript is now with our production department. 

Kind regards, 

on behalf of

Dr Daniel Pang 

Academic Editor

PLOS ONE